

# A multi-level canopy radiative transfer scheme for ORCHIDEE (SVN r2566), based on a domain-averaged structure factor

Matthew J. McGrath[1,*], James Ryder[1,*], Bernard Pinty[2], Juliane Otto[1,3], Kim Naudts[1,4], Aude Valade[5], Yiying Chen[1,6], James Weedon[7], and Sebastiaan Luyssaert[1,8]

[1]Laboratoire des Sciences du Climat et de l'Environnement, LSCE/IPSL, CEA-CNRS-UVSQ, Université Paris-Saclay, 91191 Gif-sur-Yvette, France
[2]Land Resources Management Unit, Institute for Environment and Sustainability, Joint Research Centre, European Commission, Ispra, Italy
[3]now at: Climate Service Center Germany, Helmholtz-Zentrum Geesthacht, 20095 Hamburg
[4]now at: Max Planck Institute for Meteorology, Hamburg, Germany
[5]Institut Pierre Simon Laplace, Place Jussieu 4, 75010 Paris, France
[6]now at: National Central University, Graduate Institute of Hydrological & Oceanic Sciences, Taoyuan City, Taiwan
[7]VU Amsterdam, Department of Ecological Science, 1081 HV Amsterdam, Netherlands
[8]now at: VU Amsterdam, Department of Ecological Science, 1081 HV Amsterdam, Netherlands
[*]Equal contributions

*Correspondence to:* James Ryder
(james@jamesryder.org)

**Abstract.** In order to better simulate heat fluxes over multilayer ecosystems, in particular tropical forests and savannahs, the next generation of Earth system models will likely include vertically-resolved vegetation structure and multi-level energy budgets. We present here a multi-level radiation transfer scheme which is capable of being used in conjunction with such methods. It is based on a previously established scheme which encapsulates the three dimensional nature of canopies, through the use of a domain-averaged structure factor, referred to here as the effective leaf area index. The fluxes are tracked throughout the canopy in an iterative fashion until they escape into the atmosphere or are absorbed by the canopy or soil; this approach explicitly includes multiple scattering between the canopy layers. A series of tests show that the results from the two-layer case are in acceptable agreement with those from the single layer, although the computational cost is necessarily increased due to the iterations. The ten-layer case is less precise, but still provides results to within an acceptable range. This new approach allows for the calculation of radiation transfer in vertically resolved vegetation canopies simulated in global circulation models.

## 1 Introduction

The simplest representation of canopy processes is given by the 'big-leaf' model, which replaces the entire canopy by a single vegetation element, creating homogenous microclimates and physiological responses regardless of the canopy profile. The main assumption behind the 'big-leaf' model is that a single value of leaf physiological properties can be found which adequately represents the entire canopy; given the non-linear response of photosynthesis to incoming light intensity (Björkmann, 1981), it is difficult to find such a value that works under all conditions (Leuning et al., 1995). In addition, leaf properties change over



the canopy profile, in particular in response to depth and, consequently, the amount of light reaching the leaf. The microclimate differs significantly across the canopy as well, and various photosynthetic parameters show different responses to temperature (De Pury and Farquhar, 1997). Leaf temperature is a significant driving factor in intra-canopy fluxes (Zhao and Qualls, 2005) and chemical processes, such as the emission of biogenic volatile organic carbons (BVOCs) (Guenther et al., 1993).

5      Accurate representations of photosynthesis and energy transfer within the canopy are therefore principal reasons for using multilayer canopy models (Ryder et al., 2016). To maintain model consistency, layer dimensions should coincide for the energy budget and photosynthetic calculations. This suggests using canopy layers which are a function of height within the canopy, as opposed to an alternative approach, such as layers based on the density of LAI (Leaf Area Index ). The same shortwave radiation absorbed for photosynthesis should be used in the calculation of the energy budget, which requires the creation of a 10   radiation transfer (RT) scheme capable of determining radiation fluxes as a function of height within the canopy.

     Radiation Transfer (RT) models, that resolve explicitly multiple canopy layers, have already been developed (Haverd et al., 2012; Carrer et al., 2013). The CanSPART multilayer model includes a gap fraction calculation that is based on geometric properties of the canopy instead of using a generic clumping factor (Haverd et al., 2012). The multi-level solution is generated by solving a matrix equation for the fluxes between each layer. Carrer et al. (2013) adopted a slightly different approach; the 15   calculation of the radiation that is transmitted to each layer (and, by extension, the radiation absorbed by each layer) is given directly as a product of radiation transmitted by all higher layers. Hanan (2001), Yanagi and Costa (2011) and Yuan et al. (2014) develop non-iterative two and three layer RT schemes which, while efficient, are not clearly extendable to an arbitrary number of layers.

     The ORCHIDEE-CAN model (Naudts et al., 2015) is a land-surface model that builds on the ORCHIDEE model (Krinner 20   et al., 2005; Bellassen et al., 2010) to simulate the biochemical and biophysical effects related to forest management. To better simulate complex canopies, it includes an optional multi-layer energy budget, which links to other model features such as the simulation of canopy growth, a mix of vegetation types with varying height characteristics on the same pixel, a carbon allocation scheme based on allometric relationships, inhomogenous horizontal distribution (such as tree clumping and canopy gaps), the calculation of leaf-layer resistances across the profile, and a new in-canopy radiation scheme. This latter aspect 25   necessitates the new albedo scheme that is described here.

     The increases in computational cost for the multi-layer albedo model could be justified if this version offers advantages over the single-level. One example is the significance of temperature gradients, which within canopies are often significant, even within short grassland stands (Qualls and Yates, 2001). Studies that involve of multi-layer models of the energy budget models - which requires a multi-layer radiation model - found that an accurate representation of in-canopy temperature profiles were 30   important for the simulation of sensible heat fluxes. In one study in particular, ignoring such gradients resulted in sensible heat fluxes that were 61% of reference measured values (Baldocchi and Wilson, 2001). Other studies (Ogée et al., 2003; Carrer et al., 2013) also found that simulations of the energy budget were improved with the simulation of vertical profiles.

     To be consistent across the model, the vertical albedo profile should ideally be calculated using the same technique as that used to parameterise gaps in the canopy, which in ORCHIDEE-CAN is through use of the Pgap model (Haverd et al., 35   2012), which assigns a statistical distribution of trees of varying heights and sizes. In addition the albedo model must not be





computationally expensive, should be flexible enough to be applied across a broad range of vegetation types, and should be suitable for implementation into a multi-level simulation, rather than a sun-shade type, in order to make use of the advantages of the profile approach that are detailed above. This has motivated the development of the radiation transfer scheme discussed here.

The rest of this paper is organised as follows. The next section explains the essentials of the RT model proposed by Pinty et al. (2006), and extends it to multiple layers. Details of the algorithm developed according to this theory are also presented. Results are given comparing the two-layer and ten-layer scheme directly to the one-layer scheme, applying parameter values which are representative of realistic environmental conditions (n.b. the 'REAL' parameter set), together with those spanning more of parameter space (n.b. the 'ALL' paramter set, which includes a known pathological case). Finally, we present the
limitations of the multi-level albedo model and summarise its performance against the single layer case.

## 2   Theory

The one-layer scheme is described in detail by Pinty et al. (2006). We here use the same notation and terms as developed in that paper, for consistency. As in that paper, the term 'background' used here refers to all elements except for the vegetation - essentially the soil layer, snow and leaf litter. Briefly, this scheme computes the absorption, transmission, and scattering of
incoming radiation by vegetation canopies by considering three interactions, as follows:

1. The first interaction is the radiation that does not collide with the canopy vegetation at all, and reflects off the background and back into the sky, with no interception by vegetation (this is 'black canopy radiation', or radiation with no contributions from canopy interactions; $R_{bgd}^{uncoll}$).

2. The second is the radiation which collides with the canopy elements, with a probability to be transmitted through,
absorbed by and scattered by the canopy with no contribution from the background (this is 'black background radiation'; $R_{veg}^{coll}$).

3. The third term consists of all radiation which collides with both the canopy and the background, before being scattered into the atmosphere ('multiple interaction terms'; $R_{bgd}^{coll}$; light reflected following multiple collisions between vegetation and background).

From here, one can consider each of the three possible fates of a radiation stream entering a given canopy layer from the top:

1. It can be transmitted through the layer without colliding with any vegetative elements ($T_{veg}^{\text{UnColl}}$).

2. It can be transmitted through the bottom of the layer after striking vegetation one or multiple times ($T_{veg}^{\text{Coll}}$).

3. It can exit through the top of the layer (effectively 'reflected') after colliding with vegetation one or multiple times ($R_{veg}^{\text{Coll}}$).





These three possibilities are illustrated in the top panel of Figure 1, and they constitute the basic step in the multi-level approach. Conceptually, this picture is directly comparable to the original model of Pinty et al. (2006) for the case of a non-reflecting background.

The extension of this approach to the multi-level case is conceptually straightforward, although the implementation requires

some modifications. A single unit of flux (originating from either a direct or diffuse radiation source) is projected into the top layer of the canopy. The probability that this unit will follow one of the three paths in the top panel of Fig. 1 is computed by solving the equations of Pinty et al. (2006) for the top layer of the canopy assuming a black background (so that at this step no radiation enters the layer from below). The estimate of the various fluxes requires knowledge about the effective values of the effective Leaf Area Index ($LAI_{eff}$), single scattering albedo ($w_l$) and scattering phase function for that particular layer. The

fraction scattered off the top layer and back into the atmosphere will not have another chance to interact with the canopy, and therefore it becomes the first approximation to the top of canopy albedo. The two fluxes which exit the bottom of the layer are used as inputs into the layer underneath, together with fluxes entering that layer from below (i.e. from the ground, in the two-layer model). The fate of these fluxes is calculated by again solving the equations of Pinty et al. (2006) for this layer, and the resulting fluxes are followed until all of the original flux has either been absorbed by the background, absorbed by the canopy,

or reflected back into the atmosphere. The background is treated as a further layer, with the difference that radiation can only be absorbed or reflected by it, and the proportion of radiation reflected is simply proportional to the background reflectance.

Given that the radiation transfer problem is solved using a two-stream solution for each individual sub-layer, our proposed scheme assumes that the exiting radiances in the upward and downwards directions can be appoximated by the exiting fluxes - i.e. directionality is not maintained. This assumption becomes especially critical for layers of intermediate density. The

intensity of the diffuse fluxes is numerically small for low vegetation density conditions (i.e. with an $LAI_{eff}$ that approaches 1), as a single scattering regime dominates.

The assignment of an effective LAI to each level is an appropriate simulation as the direct transmission values of the light transmitted can be calculated as the product of the layers concerned. However, we make the assumption that other factors, such as the single scattering albedo, can be assigned directly. In fact these values will be affected by back-scattering and diffuse

transmission of light between layers, and so depend slightly on the effective LAI at each level. A refinement of the model, beyond the scope for this paper, would be to run a series of convergence tests to determine more accurately the single-scatter albedo for each level.

One important consideration in the original scheme of Pinty et al. (2006) is that of the radiation that is transmitted through the canopy originating from a diffuse source, without colliding with the vegetation. Several versions of this formulation were

given (Eqns. 16, 18, and 19 in Pinty et al. (2006), taking into account a small error in the original version of Eqn. 16). The most accurate solution is Eqn. 16 from Pinty et al. (2006):

$$\overline{T_{\text{veg}}^{\text{UnColl}}} = \exp\{-LAI_{eff}/2\}\left(1 - (LAI_{eff}/2) + (LAI_{eff}/2)^2 \times \Gamma(0, (LAI_{eff}/2))\right) \tag{1}$$

(Refer to Table 1 for a summary of the abbreviations)





Given the inclusion of the incomplete gamma function, this equation is computationally somewhat expensive to solve. Therefore Pinty et al. (2006) proposed two approximate solutions, as well, noting that the following works fairly well for typical values of the effective LAI:

$$\overline{T_{\text{veg}}^{\text{UnColl}}} = \exp\{-LAI_{eff}/2\}\left(\frac{1}{1+LAI_{eff}/2}\right) \tag{2}$$

Assuming the argument of the exponential is small enough, even this approximation can be further simplified, weighting by an empirical factor:

$$\overline{T_{\text{veg}}^{\text{UnColl}}} = \exp\{-0.705 * LAI_{eff}\} \tag{3}$$

The impact of these three equations for the un-collided isotropic radiation (that is to say, of uniform magnitude in all directions) on the multi-level solution will be explored in more detail in the following section.

## 3  Algorithm

As can be understood from Figure 1, the number of fluxes increases exponentially with each step. It was anticipated that in truly pathological cases (i.e. those with a highly reflective background and non-absorbing canopy elements) the scheme would take dozens of steps to converge. In order to avoid the difficulty of tracking each flux at every step, the fluxes were combined. It is clear from the theoretical description that only two variables determine how a given flux will interact with the canopy: the direction it is travelling, and if the flux is collimated (i.e. composed of consistently parallel beams) or isotropic. Furthermore, since we assume our background is Lambertian (that is to say, diffusely reflecting) and our canopy elements are bi-Lambertian (Lambertian for both transmitted and reflected light), we do not have to track an upward collimated flux; any radiation travelling in the upward direction is necessarily scattered, and therefore will be diffuse according to our assumptions of the scattering elements. This results in only three fluxes to track for each layer, as illustrated - $T_{coll}$ (transmitted collided), $T_{uncoll}$ (transmitted uncollided) and $R_{coll}$ (reflected uncollided).

The algorithm is outlined below:

1. Use the one-layer model of Pinty et al. (2006) to compute the fraction of radiation which is reflected, transmitted after scattering, and transmitted without interacting with the canopy for each layer for both collimated and isotropic radiation sources, assuming an input flux of unity. These are referred to as the "unscaled" fluxes.

2. Initialise all the fluxes. For a collimated radiation source, the atmospheric collimated downwelling flux is set equal to unity. For an isotropic radiation source, the atmospheric isotropic downwelling flux is also set equal to unity.

3. Begin the convergence loop.

4. Initialise the variables which track the fluxes generated by this step (referred as the fluxes for the next step).

5. Start the loop over all levels in the system.



6. For each layer, determine the fraction of collimated downwelling radiation for the layer which is converted into down-welling collimated radiation for the lower layer (no interaction with the canopy), downwelling isotropic radiation for the lower layer (forward scattering by canopy elements), and upwelling isotropic radiation for the upper layer (back scattering by canopy elements). This consists of multiplying the current flux for the layer by the unscaled fluxes computed above.

7. Determine the fraction of isotropic downwelling radiation for this layer which is converted into downwelling isotropic radiation for the lower layer (no interaction with the canopy), downwelling isotropic radiation for the lower layer (forward scattering by canopy elements), and upwelling isotropic radiation for the upper layer (back scattering by canopy elements). Note that no downwelling collimated radiation can be produced by this step.

8. Determine the fraction of isotropic upwelling radiation for this layer which is converted into upwelling isotropic radiation for the upper layer (no interaction with the canopy), upwelling isotropic radiation for the upper layer (forward scattering by canopy elements), and downwelling isotropic radiation for the upper layer (back scattering by canopy elements). Note that no downwelling collimated radiation can be produced by this step.

9. Any downwelling collimated or isotropic radiation which reaches the background can become upwelling isotropic radiation for the next step by reflecting off it. The background reflectance is a fixed parameter, as in the single layer solution of Pinty et al. (2006).

10. Convergence is satisfied when all fluxes at this step are less than a specified threshold.

From this algorithm, it is clear that the total top of the canopy albedo is simply the sum of all the upwelling isotropic radiation in the atmospheric layer over all the iteration steps. One can compute the total amount of radiation reaching the background in a similar manner, as well as the fluxes within the canopy. The absorption of radiation by each canopy layer is calculated by taking the difference of the incoming radiation to each layer (from above and below) and the outgoing radiation in each direction. For the top layer, this is given by:

$$A_{veg,z=z_{top}} = 1 + R^{coll}_{veg,z=z_{bottom}} - (R^{coll}_{veg,z=z_{top}} + T^{tot}_{veg,z=z_{top}}) \qquad (4)$$

where $R^{(coll)}_{veg,z=z_{bottom}}$ is the reflected radiation from the bottom layer (the sum of all upwelling fluxes), $R^{(coll)}_{veg,z=z_{top}}$ is the reflected radiation from the top layer (the top of the canopy albedo), and $T^{tot}_{veg,z=z_{top}}$ is the total radiation transmitted through the top layer (the sum of all downwelling fluxes from the top layer).

Similarly, the absorption of the bottom canopy layer is given by:

$$A_{veg,z=z_{bottom}} = 1 + T^{tot}_{veg,z=z_{bottom}} * R_{bgd} - (R^{coll}_{veg,z=z_{bottom}} + T^{tot}_{veg,z=z_{bottom}}) \qquad (5)$$

where $T^{tot}_{veg,z=z_{bottom}}$ is the total radiation transmitted through the bottom layer, $R_{bgd}$ is the background reflectance $R^{coll}_{veg,z=z_{bottom}}$ is the total upwelling radiation from the bottom layer, and $T^{tot}_{veg,z=z_{bottom}}$ is the total downwelling radiation from the bottom layer.



The multi-level case requires an iteration scheme yielding an update of the upper and lower boundary conditions associated with each layer until reaching the appropriate flux balance.

## 4   Validation

This new albedo simulation has been written as part of a new albedo module in ORCHIDEE-CAN (Naudts et al., 2015), and
integrates with the multi-level calculations of stomatal conductance, and the multi-level energy budget. For this set of tests, the model is run with six independent variables, that are fed directly into the albedo routine in order to access the capability of this scheme.

Five of the variables also influence the single-layer scheme: the effective values of the total leaf area index (LAI), the single scatterer albedo ($w_1$), the forward scattering efficiency ($d_1$), the solar zenith angle (SZA - n.b. a value of $0°$ corresponds to the
sun directly overhead), and the background reflectance. In addition to the single layer case, we must also look at the distribution of the LAI between the two canopy layers. For simplicity, we will use the terms 'LAI' and 'effective LAI' interchangeably below in discussing the sensitivity studies, as we are not using LAI from data and thus have no reason to convert to the effective LAI required by the model. As six independent variables is too many to perform an exhaustive sensitivity analysis, we selected just two parameter sets. One parameter set, denoted ALL, covers a wide range of possible values of the parameters. The second
set, denoted REAL, focuses on a range of values which are more likely to be observed in nature. REAL is a subset of ALL, comprising almost an order of magnitude fewer points. The specific values used are given in Table 2.

For each possible combination of parameters given in Table 2, we computed the single- and two-layer solutions. The single layer case has been extensively validated (Pinty et al., 2006), and therefore we consider it to be a good reference case - that is to say, if the multi-level results match the single layer results, they can be considered as acceptable for further applications.
There are four major output fluxes of interest: the top of the canopy albedo ($R^{coll}_{veg,\,z=z_{top}}$), the total transmission through the canopy ($T^{total}_{veg}$), the total absorption by the canopy ($A^{total}_{veg}$), and the total absorption by the background ($A^{total}_{background}$). All of these fluxes are present in both the single and multi-level schemes, which makes them easy to compare, even if slightly more work is required in the multi-level case. The total canopy absorption is given by:

$$A^{total}_{veg} = A_{veg,\,z=z_{top}} + A_{veg,\,z=z_{bottom}} \tag{6}$$

while the absorption of the background is whatever radiation is not reflected as albedo or absorbed by the canopy:

$$A^{total}_{background} = 1 - (A^{tot}_{veg} + R^{coll}_{veg,\,z=z_{top}}) \tag{7}$$

In order to investigate whether some parameter settings are more prone to deviations than others, an approach of Generalised Additive ModelS (GAMS, Hastie and Tibshirani (1990)) was applied to both sets of model output - that is to say across the REAL and the ALL value range. The approach calculates the extent to which the variance of each of the two dependent
variables can be explained by each of the input terms in the model.

First of all, the full model variance is calculated for each of the output (that is to say dependent) variables - in terms of all independent input variables, both for the first order, and for the second order, tensor, interactions - that is to say the interactions



of these variables with each other. Next, the full model variance calculation is conducted again, but with one term excluded - this calculation is repeated for each of the first order and second order terms. The difference between the full model variance and the variance with one term excluded, as a fraction of the null model variance provides a value for the contribution to the variance from each term.

## 5   Results

The four main fluxes of the two-layer case agree well with those for the single layer case for the ALL parameter set (Figure 2). The results are encouraging as there seem to be no obvious cases where the multi-level approach fails, although deviations occur more frequently for fluxes which are not extreme (0.2–0.4). The resolution of this analysis is not high enough to identify more systematic differences, however, so the difference between the one- and two-layer fluxes were plotted as a function of the one layer flux, for the ALL parameter set (Figure 3). The difference between the one- and two-layer fluxes depend strongly on the category of flux of interest. The two-layer flux for the transmission through the canopy is larger than that in the one-layer case. The magnitude of the difference can reach 0.04, although most values are well below that. A small fraction of the differences between the two-level and the one-level case may be related to the assumption of an assigned effective single scattering albedo to each level in the model, as discussed in Section 2. These observations for the ALL parameter set appear to be also true for the REAL parameter sets (Figure S2).

As the fraction of LAI found in each layer cannot be specified in the single-layer case, it is instructive to see how the variation of this quantity effects the agreement between the two models (Figure 4), for the ALL parameter set. The magnitude of the differences is identical to that in Figure 3, which is to be expected as the data sets are identical. It is also unsurprising that when all of the LAI is in either the upper or lower layer the two-layer fluxes match the one-layer fluxes; in these cases, the model reduces to the single layer version. One interesting observation from Figure 4 is that the differences are asymmetric - a 1:9 distribution of top:bottom LAI gives greater differences than a 9:1 distribution. This suggests that the initial scattering by the top layer is more influential than multiple-scattering between the background and bottom layer. Again, these observations for the ALL parameter set appear to be also true for the REAL parameter sets (Figure S3).

Although the previous figures (Figures 2, 3, and 4) give a satisfactory pictorial overview of the model performance, they do not provide a quantitative assessement. To quantify model performance, we computed the fraction of simulations for which the difference between the one- and the two- layer case, not distinguishing between fluxes, was larger than a specified threshold (Table 3). These results display some expected behaviour, in particular that the fraction of simulations considered 'divergent' decreases as the threshold increases; fewer than 15% of points differ by an absolute value of more than 0.01. This value was chosen to be an 'acceptable' deviation, in the sense that a global simulation with values for individual pixels varying by 0.01 units from observed data is not considered cause for concern.

At first glance, it may seem strange in Table 3 that the numbers are higher for the REAL parameter set than the ALL parameter set , as REAL should be a subset of ALL. The reason for this is that the numbers give the fraction of points greater than a threshold, not the total number of points. For example, ALL contains 224,575 simulations which differ by more than



0.01 from the single-layer solution, while REAL has only 43,800. This is expected as REAL is a subset of ALL. However, ALL has 2,594,592 total simulations while REAL only has 304,128. Therefore, the fraction of divergent points is clearly greater. *A priori* one might expect the deviations to occur for the extreme values, *i.e.* in the ALL set but not the REAL set, an issue we will return to at the end of this section.

One natural question arising from Table 3 regards the simulations that differ by more than 0.01. Is there a common trend there which can be identified? To identify possible trends, we isolated all such simulations and applied a Generalised Additive Models (GAMS). Figure 5 depicts the resulting calculated deviances for ($\alpha_{collim}^{multi}$ - $\alpha_{collim}^{single}$), which is the difference in calculated collimated albedo between the multi-layer and single layer model. The analysis reveals that, for the ALL parameter set, medium values of LAI, small forward scattering efficiencies, evenly distributed vegetation, and high single scatterer albedos all lead

to increased frequencies of points which deviate significantly from the single layer model. The solar zenith angle and the background reflectance appear to have little effect on the differences between the one- and two- layer models. Comparison between the ALL parameter set (Figure 5) and the REAL (not shown) reveals that the REAL parameter sets results in higher deviations than the ALL settings, reiterating that the multi-level model appears more well-behaved for extreme parameter values.

The complete model is run over two layers, as in the previous tests, and also over ten layers, in a similar experimental setting to that of Chen et al. (2015). We found that the most significant contribution to the variance for both the ten- and two- layer cases comes from $w_l$, the single scattering albedo. Figure S4, plots the marginal effects, that is to say, the effects of each of the independent input values against the output variable (which is in this case the outcome of the flux difference threshold test). Note that for a threshold of 0.01, the ten- layer case requires almost 10 times as many simulations than the two layer case to

fail the threshold. The number of simulations that fail the threshold for the ten layer case equal the number of simulations of the two layer case when the threshold is increased from 0.01 to 0.03.

     From a computational point of view, it is clear that the iterative procedure introduced above will be more demanding of computer resources than the original one-layer scheme. Two valid questions are therefore how much more expensive it can be, and under what conditions will this expense be increased? To answer these, we have applied a GAMS to the mean number of

iteration steps ($n_{steps}$) to convergence. The partitioning of the LAI into the layers also has a significant effect on the expense of the algorithm, requiring more time as the bottom layer becomes empty. The single scattering albedo ($w_l$), is the dominant input variable, followed this time by interactions between that variable and $R_{bgd}$, the collimated background reflectance, as shown in Figure 6. The solar zenith angle is not an important factor, resulting in only small differences. These analyses show that as more light is scattered, for example, with a more reflective background or leaves, the number of steps required to converge

can increase by a factor of five (shown in Figure S5). Finally, Figure S6 plots the marginal effects for a difference in albedo between the two- and one- layer model, for the both the REAL and ALL datasets. Figure S7 is the corresponding graph for the number of iteration steps ($n_{steps}$), which shows little difference between the two value tests for this metric.





## 6  Discussion

As the multi-level radiation transfer model is designed to run at a global scale, it is important to assess the scheme in regions where changes in albedo have the most consequence for the global climate. For example in the northern latitudes albedo has a key effect in the springtime, when solar angles are increasing and there is still a broad snow cover. In these circumstances, the

results (Figure S4(e)) show that as the solar zenith angle reaches a large value, the difference between the calculated albedo, both for a single and the multi-level version, decreases and so the accuracy improves with the seasonal variation in radiation. In absolute terms, the calculated albedo has the greatest effect in the tropics, as the largest amount of direct shortwave radiation falls here in this region. Of course, the solar zenith angle will have a higher mean value towards the lower end of the range displayed in figure S4e). So deviations in absolute terms are expected to be relatively low due to the high LAI in the tropics.

The interactions between leaf irradiance in different parts of the canopy and water stress have an impact on $CO_2$ concentration within these leaves. When using a multi-layer canopy model, errors in Carbon Gross Primary Productivity (GPP) were considerably reduced, as a result of improved radiation distribution (Bonan et al., 2012) and, in a later work, also by improved leaf moisture gradients (Bonan et al., 2014), when compared to a sun-shade model. As a next step, the development of models such as these enables a consistent approach to canopy modelling that can link energy budget and $CO_2$ models with other

DGVM (Dynamic General Vegetation Model) processes.

There has recently been an improvement in measurement capability within canopies, such as the development of a more portable LIDAR (Hosoi and Omasa, 2007), the increased availability (albeit gradually) of more detailed in-canopy datasets of canopy structure and the improved coordination and collation of time series measurements of in-canopy temperature, humidity and trace gases (Chen et al., 2015; Sellers et al., 1997). Following this trend in data availability, the most recent generation of

DGVM models include the simulation of canopy growth, vertical canopy profile information and leaf level resistances, such as stomatal conductance (e.g. in ORCHIDEE-CAN (Naudts et al., 2015) and the CLM (Community Land Model) (Bonan et al., 2012, 2014)). This means that profile based models will increasingly couple not only with the atmosphere, but with other modules that provide more detail of canopy composition.

## 7  Conclusions

We have developed an algorithm to extend a powerful single-layer canopy radiative transfer model to multiple layers. The original radiative transfer scheme incorporated three dimensional canopy structure through the consideration of a domain-averaged structure factor, providing the two-stream fluxes as output. Our extension here tracks the fluxes as they pass through the canopy layers, using an iterative procedure, until they escape into the atmosphere or the background; in this way, multiple scattering between the canopy layers is taken into account, as well as the multiple scattering between the canopy and the

background included in the single-layer scheme.

Despite the fact that computational cost increased and divergence of the multi- compared against the one- layer model occurs, especially for realistic parameter values, the magnitude and sensitivity of the divergence does not hamper use of the model for global simulations.





The results of the tests presented here are encouraging, showing overall good agreement between the one- and two-layer models; deviations are no more than 0.04 albedo units. While this difference is not insignificant, deviations are typically below 0.01. Some parameters (primarily $w_l$, the leaf single scatterer albedo, but also $R_{bgd}$, the background reflectance) appeared to have larger effects on the agreement between the two approaches than others. The computational cost of the multi-level

5  approach was also examined. Again, some parameters (such as the background reflectance and the single scatterer albedo) had a larger impact on the number of steps required to converge to the iterative solution than others. The number of steps required to converge was only loosely correlated to increased differences between the one- and two-layer results. These results indicate that, while systems with highly reflecting background (like snow) and high single scatterer values (for example, in the infrared band) may lead to large differences with the single-layer case, overall this scheme is robust and serves as a powerful start for

10  the next generation of global land-surface models with multi-level energy budgets and three-dimensional vegetation structure.

## 8   Code availability

The code and the run environment are open source (http://forge.ipsl.jussieu.fr/orchidee). Nevertheless readers interested in running ORCHIDEE-CAN are encouraged to contact the corresponding author for full details and latest bug fixes.

*Acknowledgements.* JR, YC, MJM, JO and SL were funded through ERC starting grant 242564 (DOFOCO), and AV was funded through

15  ADEME (BiCaFF). ESA CCI Landcover also supported this work.



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





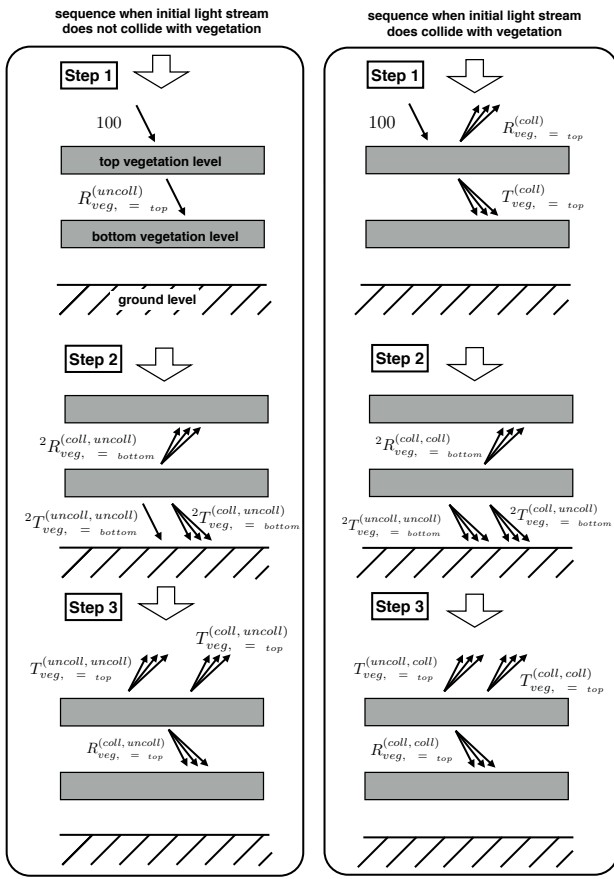

**Figure 1.** Schematic representation of the first three steps in the multi-level algorithm. The left side of the figure represents the first three steps of the situation where the initial light stream does not collide with vegetation. The right side of the figure represents the first three steps of the situation where the initial light stream does collide with the vegetation. 'T' and 'R' represent packets of light which are transmitted and reflected, respectively. 'coll' represents light that has collided with vegetation elements in the present timestep and 'uncoll' represents light that has not collided with vegetation. 'coll, uncoll' for example represents light uncollided with vegetation in the previous step, that has subsequently collided with vegetation in the present step, and so on for 'coll, coll', 'uncoll, coll' and 'uncoll, uncoll'.





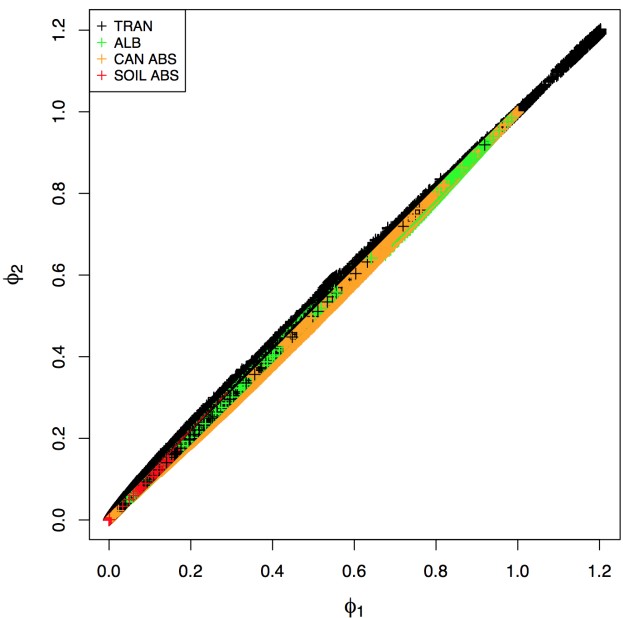

**Figure 2.** The fluxes of the the total transmission through the canopy (TRAN, or $T_{veg}^{total}$ in text), top of the canopy albedo (ALB, or $R_{veg,\ z=z_{top}}^{coll}$ in text), the total absorption by the canopy (CAN ABS, or $A_{veg}^{total}$ in text), and the total absorption by the background - that is to say soil, snow and leaf litter (SOIL ABS, or $A_{background}^{total}$). Figure is for the two layer model ($\phi_2$) as a function of the one layer corresponding results ($\phi_1$) for a broad wide range of input parameters (see table 2). Different fluxes are represented by different colours. This figure corresponds to the ALL dataset.





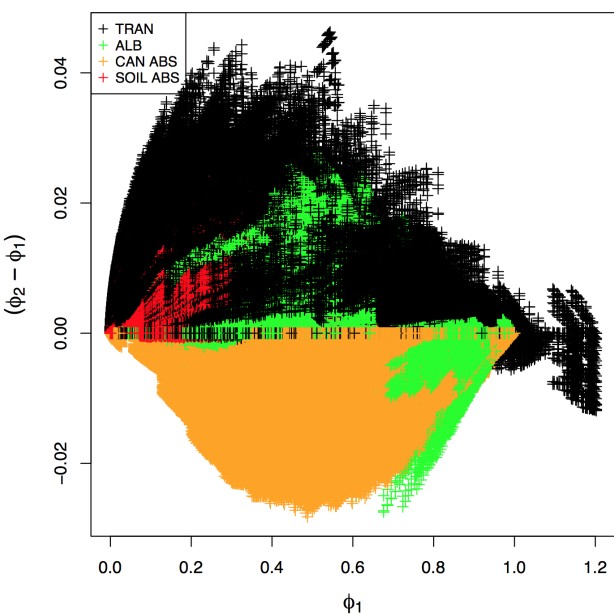

**Figure 3.** The fluxes of the the total transmission through the canopy (TRAN, or $T_{veg}^{total}$ in text), top of the canopy albedo (ALB, or $R_{veg,\ z=z_{top}}^{coll}$ in text), the total absorption by the canopy (CAN ABS, or $A_{veg}^{total}$ in text), and the total absorption by the background - that is to say soil, snow and leaf litter (SOIL ABS, or $A_{background}^{total}$). Figure shows the signed difference between the two layer results and the one layer results ($\phi_2 - \phi_1$) as a function of the one layer results for a broad range of input parameters. This figure corresponds to the ALL dataset.





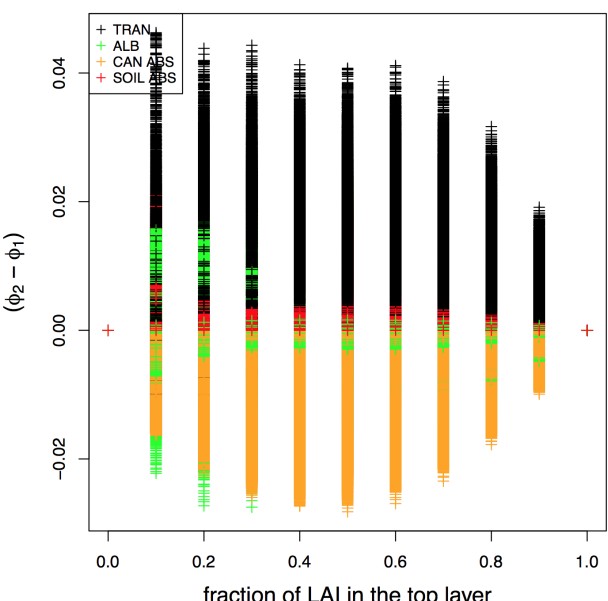

**Figure 4.** The fluxes of the the total transmission through the canopy (TRAN, or $T_{veg}^{total}$ in text), top of the canopy albedo (ALB, or $R_{veg,\ z=z_{top}}^{coll}$ in text), the total absorption by the canopy (CAN ABS, or $A_{veg}^{total}$ in text), and the total absorption by the background - that is to say soil, snow and leaf litter (SOIL ABS, or $A_{background}^{total}$). Figure is for the difference in fluxes between the two-layer and the single-layer results ($\phi_2 - \phi_1$) as a function of the fraction of LAI in the top layer for a broad range of input parameters. This figure corresponds to the ALL dataset.



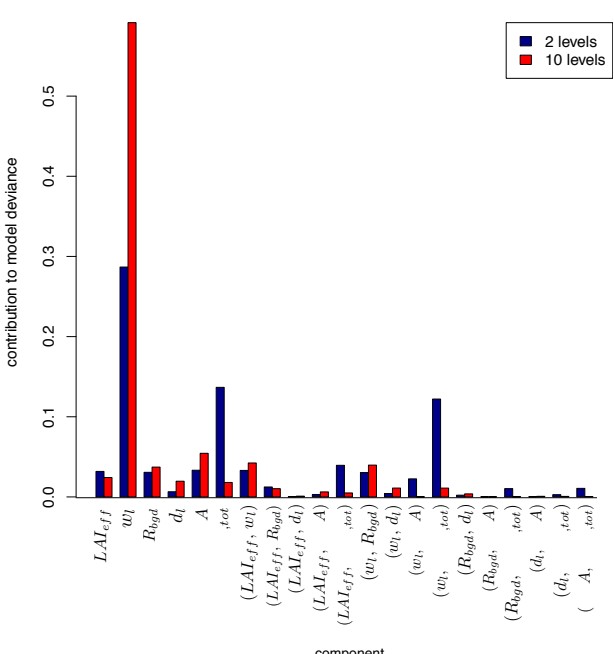

**Figure 5.** Contribution of each input variable to the deviance in the value of $(\alpha_{collim}^{multi} - \alpha_{collim}^{single})$, the difference of the collimated albedo for the multi-level (two layer version in blue and ten layer in red) and for the single layer model





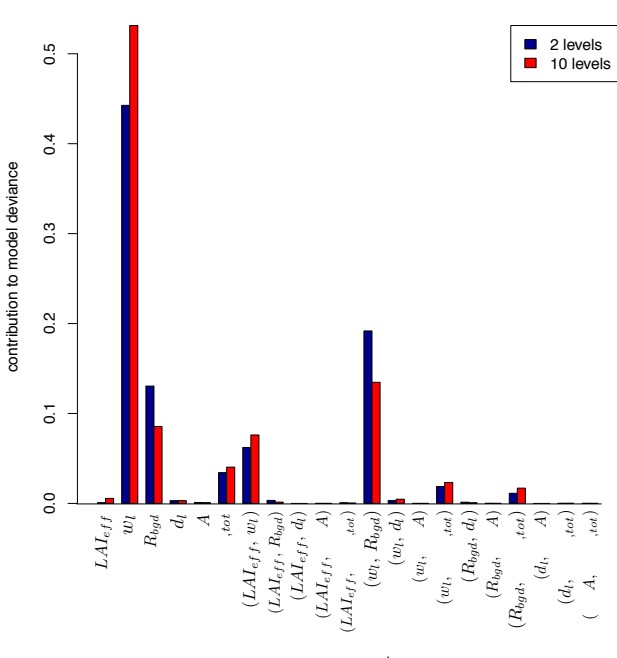

**Figure 6.** Contribution of each input variable to the deviance in the value of $n_{steps}$, the total number of iterations required to reach a result in the multi-level model (two layers in blue and ten layers in red).



**Table 1.** Key to abbreviations

| Description | Term used in code | Symbol in text |
| --- | --- | --- |
| Independent input variables to model | | |
| collimated effective Leaf Area Index (LAI) | ilaieff | $LAI_{eff}$ |
| fractional effective LAI | laieff_frac | $LAI_{eff}^{frac}$ |
| leaf single scattering albedo | iwl | $w_l$ |
| background reflectance | irbgd | $R_{bgd}$ |
| leaf forward scattering efficiency | idl | $d_l$ |
| solar zenith angle | itheta | $SZA$ |
| fraction of LAI contained within top half of the canopy | isplit | $f_{LAI,tot}$ |
| Independent output variables to model | | |
| collimated albedo for a single layer | collim_alb_tot_1 | $\alpha_{collim}^{single}$ |
| collimated albedo for total layers | collim_alb_tot | $\alpha_{collim}^{multi}$ |
| number of model iterations | nsteps | $n_{steps}$ |
| Other abbreviations for light packets | | |
| reflected stream; collided with background; uncollided with vegetation | * | $R_{veg}^{uncoll}$ |
| reflected stream; uncollided with background; collided with vegetation | * | $R_{veg}^{coll}$ |
| reflected stream; collided with background; collided with vegetation | * | $R_{bkg}^{coll}$ |
| transmitted stream; uncollided with vegetation | * | $T_{veg}^{uncoll}$ |
| transmitted stream; collided with vegetation | * | $T_{veg}^{coll}$ |
| top canopy level | | $z_{top}$ |
| bottom canopy level | | $z_{bottom}$ |
| range of input values encompassing a wide range of parameters | | ALL |
| range of input values encompassing a smaller, more realistic, range of input parameters | | REAL |

* calculated as sum of components in code (no direct term)



**Table 2.** The range of values used for the ALL test case (top) and the REAL test case (bottom). ISO indicates an isotropic view angle, i.e. integrated over all view angles as described in Pinty (2006). $LAI$ is in units of leaf area per square meter of land, $SZA$ is the solar zenith angle in degrees, and the rest of the variables are unitless.

| Variable | Values |
|---|---|
| $LAI$ | (1.0, 2.0, 3.0, 4.0, 5.0, 6.0, 7.0, 9.0, 11.0) |
| $f_{\mathrm{LAI,top}}$ | (0.0, 0.1, 0.2, 0.3, 0.4, 0.5, 0.6, 0.7, 0.8, 0.9, 1.0) |
| $d_{\mathrm{l}}$ | (0.1, 0.5, 1.0, 1.5, 2.0, 5.0, 10.0) |
| $SZA$ | (0.01, 20.0, 40.0, 60.0, 80.0, ISO) |
| $w_{\mathrm{l}}$ | (0.001, 0.05, 0.1, 0.15, 0.2, 0.3, 0.4, 0.5, 0.6, 0.7, 0.8, 0.9, 0.999) |
| $R_{\mathrm{bgd}}$ | (0.001, 0.05, 0.1, 0.15, 0.2, 0.3, 0.4, 0.5, 0.6, 0.7, 0.8, 0.9, 0.999 ) |
| $LAI$ | (1.0, 2.0, 3.0, 4.0, 5.0, 6.0) |
| $f_{\mathrm{LAI,top}}$ | (0.0, 0.1, 0.2, 0.3, 0.4, 0.5, 0.6, 0.7, 0.8, 0.9, 1.0) |
| $d_{\mathrm{l}}$ | (1.0, 1.5, 2.0) |
| $SZA$ | (0.01, 20.0, 40.0, 60.0, 80.0, ISO) |
| $w_{\mathrm{l}}$ | (0.05, 0.1, 0.15, 0.2, 0.6, 0.7, 0.8, 0.9) |
| $R_{\mathrm{bgd}}$ | (0.05, 0.1, 0.15, 0.2, 0.6, 0.7, 0.8, 0.9 ) |





**Table 3.** The fraction of data points for each combination of layer and parameter set for which the absolute value of the difference between the one-layer and $N$-layer case is greater than the specified threshold. Different equations are used for the transmission factor of uncollided isotropic radiation, as described in Section 2.

| $N_{\text{layer}}$ | Parameter set | Errors | | | | |
|---|---|---|---|---|---|---|
| | | 0.001 | 0.002 | 0.005 | 0.01 | 0.02 |
| | | Eq. 1 | | | | |
| 2 | ALL | 0.432 | 0.334 | 0.195 | 0.103 | 0.041 |
| 2 | REAL | 0.489 | 0.380 | 0.234 | 0.117 | 0.034 |
| | | Eq. 2 | | | | |
| 2 | ALL | 0.390 | 0.306 | 0.195 | 0.123 | 0.068 |
| 2 | REAL | 0.457 | 0.378 | 0.251 | 0.145 | 0.063 |
| | | Eq. 3 | | | | |
| 2 | ALL | 0.461 | 0.359 | 0.200 | 0.084 | 0.010 |
| 2 | REAL | 0.567 | 0.437 | 0.267 | 0.144 | 0.018 |