# Peer review of "A multi-level canopy radiative transfer scheme for ORCHIDEE (SVN r2566), based on a domain-averaged structure factor"

_Geoscientific Model Development, 2016_

## Referee Comment (RC1) · Anonymous Referee #1 · 8 Dec 2016

This paper presents an adaptation of an existing 1D RT model developed for large scale parameter retrieval, and modifies it to include vertical canopy layering for improved RT treatment in land surface modelling. The paper is clear and well-written and presents a practical, incremental advance of the application of the Pinty et al 2 stream approach to multi-layer canopies. There are a few general issues which could be addressed to improve the paper, which I discuss below, followed by some minor technical and grammatical comments. If these are addressed I think the paper can be published.

General comments. The analysis is presented in many cases in a rather quantitative way - terms like 'good', 'reasonable' etc are used. What is needed is a statement at the outset of what might be considered to be a good enough agreement between the

1 and multi-layer case for example. How good is good enough from the perspective of the LSM(s) into which such an RT scheme might be embedded? This might affect how likely you would be to think about incorporating this approach, as well as how many layers you might decide to use. This latter point is also not addressed - why would I use 2 layers, or 5, or 10? What would determine how many layers I might use?

So, see p8, line 25: this is crucial - where does your definition of what's acceptable come from and why is this acceptable? All the model results in the paper are essentially a function of this, or have to be interpreted in this light. So this needs clear definition and justification at the outset. Given the likely applications for the LSM with the RT embedded, what is the divergence threshold that would preclude the use of the LSM with the embedded multi-stream RT model?

There are a lot of results (like S5 for eg) which are useful, but are in Supp Info. I'm not sure about the balance here of material in the main paper v suppl. - maybe too much in the latter. Could consider this.

Regarding some of the key figures - see the comment on p8, line 24: "Although the previous figures (Figures 2, 3, and 4) give a satisfactory pictorial overview ..." no, they don't! The choice of the same symbol with no transparency for all fluxes means they cover each other and as a result the figures are too hard to interpret properly. We have to take on trust some of the discussion of these results as we certainly can't see all points for all fluxes. These results need to be presented much better- smaller points, different symbols etc.

Technical/minor comments p1, 8-9: acceptable? Can this be quantified properly?

p2, 6: multilayer canopy models *within* larger LSMs?

p2, 11: well yes, there are a lot of very detailed RT models - I think you mean here RT model schemes embedded in larger LSMs? If so, be explicit.

p2, 28: garbled sentence: "Studies that involve of multi-layer models of the energy

budget models..." - whole sentence needs re-writing.

p2, 33: "To be consistent across the model," not clear what that means here - it's sort of implicit but this issue of structural and radiometric consistency (which is what I think is meant) is important, and should be clarified.

p3, 1: but what is 'expensive' - compared to what?

p3, 8: RT parameter values?

p4, 9: the effective LAI term hasn't been described properly yet (or leaf sscatt albedo - although that is well-defined anyway). Best to define this at the outset of 2 when describing Pinty model, as this is the key parameter of the scheme. Might also be worth introducing the other parameters for the single layer scheme i.e. from p7, lines 9-10 d1, sza, and soil refl.

p4, 22: appropriate assumption?

p4, 27: single-scattered (and elsewhere in the text)

p5, 6: need to say how the empirical factor is/was arrived at.

p7, 11: isn't it simpler *just* to use LAI_eff from here on then, unless you really mean actual LAI?

p7, 25: 'as albedo'?

p8, 6 (and elsewhere): avoid use of subjective words like 'well' unless you have defined. So either define what you mean (in terms of RMSE for eg) or, better, just give the RMSE and/or r2. So for fig2, would be good to know what cases of largest departure are.

p8, 14: " a small fraction ... may be ...." - well, can you test that? If not, why?

p8, 31-and on: so why not express it in a different way?

p9,9: leaf single-scattering albedos. This is probably not surprising is it, in that these are the cases where the layers, and veg properties within each layer, have the most

impact? This is reinforced in the next para when we see that the leaf single-scattering albedo is the biggest determinant - this controls the amount and direction of the fluxes between layers.

p9, 30: number of iterations to converge - does this scale linearly with number of layers? Results in Figure S5 suggest this might be so - so couldn't you calculate that analytically? Or at the least show a case with a moderate number of layers eg 5.

p11, 1: 'good agreement' - meaning what? You give the figure (0.04 albedo) so just leave it at that unless you define at the outset what you consider to be 'good', 'acceptable', 'poor' etc.

p11, 2: '...deviations are typically...' - give a number, what fraction, for 2 and 10 layer cases?

Suppl Info

Caption of Table 1 - ontents. And also missing figure numbers.
* * *

---

## Referee Comment (RC2) · Anonymous Referee #2 · 5 Feb 2017

GMDD Review - McGrath et al., 2017: "A multi-level canopy radiative transfer scheme for ORCHID (SVN r2566), based on a domain-averaged structure factor

The authors present an evaluation of a 1-D radiative transfer scheme adapted to consider multiple levels within a vegetation canopy. The scheme is embedded in the OR-CHIDEE land surface model and could thus be included in coupled environmental and earth system models. The paper outlines the changes made to the existing single layer (or big leaf) approach and a comparison of results for 2 and 10 canopy levels vs. the original 1 layer. While the adaptations made to the scheme are much needed, although somewhat incremental, the method of evaluation and the presentation of results leave much to be desired and I feel fundamental revisions are required before the paper is

suitable for publication.

My chief concerns lie in the choice of the previous (single layer) model as truth. The authors evaluate the performance of the multi-level radiative transfer scheme by comparison against output from the current single level scheme. The skill of the current scheme is described as "good"; while the reader is referred to previous work in which this model was fully evaluated, there is no further information supplied here as to just how good that might be, nor the environments and canopy types under which it performs particularly well or poorly. We are therefore asked to judge whether or not the new scheme is an improvement on the old against an arbitrary baseline. If the new model deviates from the old by (say) 4% we have no means to determine whether that is in fact a degradation in performance or whether that change in output actually brings the new scheme in better agreement with observations. Given that the original scheme has been rigorously evaluated there is no reason that the new scheme should not be similarly compared against measurement data from a range of vegetation and environmental conditions. Without such comparison any analysis of model performance is by necessity incomplete and inadequate.

The evaluation lacks quantitative rigour, with comparisons (often referred to as "deviations") described qualitatively ("good", "reasonable", "acceptable") rather than in terms of RMSE or even percentage error. The authors do not make clear what constitutes an "acceptable" performance in the context of radiation absorbed or reflected by vegetation, and yet albedo is a key parameter in land surface and Earth system models; small changes can profoundly alter local climate and meteorology.

In addition, the skill of the new multi-level scheme to capture successfully the absorption and scattering of radiation entering the canopy should be determined separately for different circumstances. The authors do attempt to include such an assessment in their discussions of the results but again this is done in an entirely qualitative, incomplete and vague manner (e.g. P9, L8-9 presents a list of values - as "medium", "high", etc (again without making clear what they mean by these arbitrary descriptions)
- that increase deviation from the single level model). It would be of enormous value to the community were the authors to identify the subset of parameter space in which an increase in model levels improves the skill of the model, the subset for which it roughly matches the performance of the single level scheme, and that for which performance is impaired (RELATIVE TO OBSERVATIONS). Such information would be invaluable for driving further development of the representation of the vegetation canopy in large-scale models - very much a neglected region of the Earth system.

The authors present results of a multiple simulation test called REAL in which all possible combinations of realistic parameter variables are considered. They then further include a test called ALL which encompasses the full sample space of REAL but also considers extreme values which would not be encountered in the real world. I am curious as to the purpose of this set of simulations which to my mind does not help assess the genuine skill of the model, and here seems to serve only to confuse the issue given that at times the more extreme conditions at first sight improves the apparent performance of the multi-level model. A revision of the manuscript should present only the REAL simulations but, as noted above, should include far greater detail of the individual conditions represented by various parameter combinations.

It is also not clear how the space is sampled. It seems that equal weighting is given to all possible values although in life none of the variables could be expected to have a uniform distribution.

Furthermore, while the authors introduce the model by stressing the urgent need to include multi-level canopies in coupled models due to substantial differences between vegetation structure and characteristics at different heights within complex canopies, their results, discussions and conclusions do not validate this claim. Instead, the reader is left questioning why the additional computational cost would be necessary. At best, the authors conclude that the multi-level model shows good agreement with the single level. If a model "improvement" shows no clear improvement over previous versions there seems no incentive to include it in coupled models given the current demands for
additional details (and computational cost) that can be shown to be justified.

Finally, the motivation, model and results are poorly presented and explained. Insufficient consideration is given to previous work in this area: many multiple level canopy models have been developed and are in use in 1D and coupled models but these are at best only given a cursory acknowledgement in the Introduction (24 references is inadequate for a paper describing an incremental advance on previous work). Many important vegetation and canopy characteristics are left undefined (what is the "effective leaf area" for example) and different terminology is used for the same parameter (diffuse and isotropic). The domain-averaged structure factor referred to in the title is not clearly derived. The authors switch from discussing radiation to fluxes. Sunshade models are never described and it is left unclear how incoming radiation is split between direct and diffuse (or indeed if it is all assumed direct until scattered in the canopy). Single scattering albedo is often instead called single scatterer albedo.

Figures 5 and 6 do not appear to be referred to in the text and to my mind far too many figures are presented as supplemental material but then discussed at length in the main text. If a figure requires more than a brief "see Fig. Sxx" it belongs in the main paper.

---

## Author Comment (AC1) · 1 Aug 2017

(please see differences marked in attached file)

This paper presents an adaptation of an existing 1D RT model developed for large scale parameter retrieval, and modifies it to include vertical canopy layering for improved RT treatment in land surface modelling. The paper is clear and well-written and presents a practical, incremental advance of the application of the Pinty et al. 2 stream approach to multi-layer canopies. There are a few general issues which could be addressed to improve the paper, which I discuss below, followed by some minor technical and grammatical comments. If these are addressed I think the paper can be published.

[Figure]

(1) General comments. The analysis is presented in many cases in a rather quantitative way - terms like 'good', 'reasonable' etc are used. What is needed is a statement at the outset of what might be considered to be a good enough agreement between the single and multi-layer case for example. How good is good enough from the perspective of the LSM(s) into which such an RT scheme might be embedded?

**The following has been added to the text (page 10, line 26):**

**For consistency when comparing results, the following set of thresholds was agreed for determining the level of agreement between the albedo that was calculated, between the single-level and the multi-level model:**

**- a difference of < 0.01 represents a minimal change, which would be equivalent to a top of canopy measurement error**

**- a difference of 0.01 - 0.05 represents an acceptable change**

**- a difference of > 0.05 represents a substantial change - anything above this approaches the differences between the albedo for different land surface types (e.g. evergreen versus deciduous forest) and so is ruled out**

(2) This might affect how likely you would be to think about incorporating this approach, as well as how many layers you might decide to use. This latter point is also not addressed - why would I use 2 layers, or 5, or 10? What would determine how many layers I might use?

**The following has been added to the text (page 9 line 22):**

**This new albedo simulation has been written as part of a new albedo module in ORCHIDEE-CAN (Naudts et al., 2015), and integrates with the multi-level calculations of stomatal conductance, and the multi-level energy budget. The specific number of layers to be used depends on the other aspects of the canopy model, particularly the energy budget and the photosynthesis scheme. It is the nature of these latter schemes that would determine the number of layers to be used. For example, an energy budget calibrated for an understorey/overstorey ecosystem may run on two layers, whilst an energy budget for a more complex canopy might use ten. The number of layers in the albedo scheme would match those required by the other processes; our scheme is sufficiently flexible that changing this in the model is straightforward. For the set of tests in this study, the model is run for 1, 2 and 10 levels.**

(3) So, see p8, line 25: this is crucial - where does your definition of what's acceptable come from and why is this acceptable? All the model results in the paper are essentially a function of this, or have to be interpreted in this light. So this needs clear definition and justification at the outset. Given the likely applications for the LSM with the RT embedded, what is the divergence threshold that would preclude the use of the LSM with the embedded multi-stream RT model?

**See addition to text in response to (1), above.**

(4) There are a lot of results (like S5 for eg) which are useful, but are in Supp. Info. I'm not sure about the balance here of material in the main paper v suppl. - maybe too much in the latter. Could consider this.

**We have moved figures S5, S6 and S7 to the main text.**

(5) Regarding some of the key figures - see the comment on p8, line 24: 'Although the previous figures (Figures 2, 3, and 4) give a satisfactory pictorial overview...' no, they don't! The choice of the same symbol with no transparency for all fluxes means they cover each other and as a result the figures are too hard to interpret properly. We have to take on trust some of the discussion of these results as we certainly can't see all points for all fluxes. These results need to be presented much better- smaller points, different symbols etc.

**Each graph replaced by combination figure of four subplots, where each sub-plot highlights one component with larger symbols (i.e. transmitted, reflected, absorbed by canopy and absorbed by soil fraction), and shows the other components with smaller, fainter symbols.**

**First part of paragraph replaced with (page 12, line 3): 'Figures 2, 3, and 4 demonstrate the visual extent of variation in the calculated albedo between the single and two-level model.'**

**Inserted further in the same paragraph (page 12, line 5): 'For the REAL dataset using the two layer model, the fluxes of the majority of test points differ by 0.01 or less compared to the well-validated single-layer model in 85% of cases, and by 0.02 or less in 98% of cases (see Table 3). There are no incidences for which the comparisons differ by more than 0.03.'**

**Technical/minor comments:**

(6) p1, 8-9: acceptable? Can this be quantified properly?

**Inserted (page 1, line 9): For the 2 level case fewer than 15% of results across the variable range have a divergence greater than 0.01, fewer than 2% greater than 0.02, and none have a divergence greater than 0.03.**

**Inserted (page 1, line 11): (40% of results across the variable range have a divergence greater than 0.01, 10% across the variable range have a divergence greater than 0.03, and 1.5% greater than 0.05)**

(7) p2, 6: multilayer canopy models *within* larger LSMs?

**Added (page 2, line 10): 'Accurate representations of photosynthesis and energy transfer within the canopy are therefore principal reasons for using multilayer canopy models both for single site assessments and for regional and global studies within large-scale land surface models.'**

(8) p2, 11: well yes, there are a lot of very detailed RT models - I think you mean here RT model schemes embedded in larger LSMs? If so, be explicit.

**Modified to read (page 3, line 22): 'Radiation Transfer (RT) models, that resolve explicitly multiple canopy layers as part of a large-scale LSM, have already been developed.'**

(9) p2, 28: garbled sentence: 'Studies that involve of multi-layer models of the energy budget models...' - whole sentence needs re-writing.

**Modified to read (page 2, line 25): 'Studies that model the energy budget using an in-canopy profile - which also requires a multi-layer radiation model - found that an accurate representation of in-canopy temperature profiles was important for the simulation of sensible heat fluxes.'**

(10) p2, 33: 'To be consistent across the model,' not clear what that means here - it's sort of implicit but this issue of structural and radiometric consistency (which is what I think is meant) is important, and should be clarified.

**Modified to read (page 4, line 19): 'To be consistent with the vertical canopy structure (that is to say, the Plant Area Density profile) that is used across several parts the model, the albedo profile should be calculated using the same technique as that used to parameterise gaps in the canopy, which in ORCHIDEE-CAN is through use of the Pgap model (Haverd, 2012b).'**

(11) p3, 1: but what is 'expensive' - compared to what?

**Modified to read (page 4, line 29): The increase in computation time should be acceptable. In state-of-the-art coupled atmospheric-land models, the atmospheric model requires about 10 times as much computer resources than the land surface model (this is approximately the ratio that applies to the ORCHIDEE model for the uncoupled versus the version coupled to the LMDZ (Laboratoire de Météorologie Dynamique Zoom) atmospheric model). The albedo model should thus not increase the computational cost by a factor of two or more to avoid the land surface becoming a rate limiting step in coupled atmospheric-land models.**

(12) p3, 8: RT parameter values?

**See details regarding performance thresholds by which the model is assessed, as in response to point (1), above.**

(13) p4, 9: the effective LAI term hasn't been described properly yet (or leaf scatt albedo - although that is well-defined anyway). Best to define this at the outset of 2 when describing Pinty model, as this is the key parameter of the scheme. Might also be worth introducing the other parameters for the single layer scheme i.e. from p7, lines 9-10 d1, sza, and soil refl.

**(page 6, line 13) List of descriptions of effective LAI, and the other key parameters, inserted here for clarity.**

(14) p4, 22: appropriate assumption?

**Corrected to 'appropriate assumption', from 'appropriate simulation'(page 7, line 3)**

(15) p4, 27: single-scattered (and elsewhere in the text)

**All instances of 'single-scatter' (and variations thereof) changed to 'single-scattering albedo'.**

(16) p5, 6: need to say how the empirical factor is/was arrived at.

**Added (page 7, line 21): '(the determination of which is explained in Section 2.5 of (Pinty et al., 2006)'**

(17) p7, 11: isn't it simpler just to use LAI_eff from here on then, unless you really mean actual LAI?

**We have replaced references to LAI with references to LAI_eff from this point. Sentence rephrased to read (page 6, line 20): 'For consistency, we will use 'LAI_eff', to also denote LAI in discussing the sensitivity studies. As we are not using LAI from data there is no reason to convert to the effective LAI required by the model.', and moved upwards to first mention of effective LAI, in the theory section.**

(18) p7, 25: 'as albedo'?

**Corrected - should refer to the 'radiation flux not reflected' (page 10, line 25)**

(19) p8, 6 (and elsewhere): avoid use of subjective words like 'well' unless you have defined. So either define what you mean (in terms of RMSE for eg) or, better, just give the RMSE and/or r2. So for fig2, would be good to know what cases of largest departure are.

**RMSE table calculated for Figure 2 (see Table 4 and references in text)**

(20) p8, 14: 'a small fraction ... may be....' - well, can you test that? If not, why?

**A small fraction of the differences between the two-level and the one-level case may be related to the assumption of an fixed effective single scattering albedo to each level in the model, as discussed in Section 2. From a model standpoint, it is possible to test this, by assigning different values to each level. The challenge is interpreting the results. By introducing additional parameters, it is reasonable to expect that we would be able to better fit the results. Without a strong physical basis on which to estimate these values, however, we run a strong risk of 'over-fitting' our model. In this context, a 'better' fit does not give us new information on the contribution of this assumption to the difference between the two-level and one-level cases. For this reason, we do not feel running these tests would create meaningful results.**

(21) p8, 31-and on: so why not express it in a different way?

**Replaced 'it may seem strange' with 'it may seem counter intuitive' (page 10, line 25)**

(22) p9,9: leaf single-scattering albedos. This is probably not surprising is it, in that these are the cases where the layers, and veg properties within each layer, have the most impact? This is reinforced in the next para when we see that the leaf single-scattering albedo is the biggest determinant - this controls the amount and direction of the fluxes between layers.

**Included additional commentary in the next paragraph to this (page 12, line 27): 'As the layers in the canopy profile have been added, this demonstrates that factors that control the passage of fluxes between layers as expected have the most significant influence on the total flux from the canopy.'**

(23) p9, 30: number of iterations to converge - does this scale linearly with number of layers? Results in Figure S5 suggest this might be so - so couldn't you calculate that analytically? Or at the least show a case with a moderate number of layers eg 5.

**This is an interesting suggestion for future work. We however cannot presently afford extra runs with the model to perform the required sets of simulations. Figure S7 was added here to simply provide an indication to the users.**

(24) p11, 1: 'good agreement' - meaning what? You give the figure (0.04 albedo) so just leave it at that unless you define at the outset what you consider to be 'good', 'acceptable', 'poor' etc.

**We have replaced qualitative judgments with the actual figures as suggested.**

(25) p11, 2: '...deviations are typically...' - give a number, what fraction, for 2 and 10 layer cases?

**Added (page 14, line 20): 'While this difference is not insignificant, fewer than 15% of divergences are greater than 0.01 in the two level case, and for the ten level case, 10% across the variable range have a divergence greater than 0.03.'**

Supplementary Info

(26) Caption of Table 1 - ontents. And also missing figure numbers.

**Corrected**

Please also note the supplement to this comment:
https://www.geosci-model-dev-discuss.net/gmd-2016-280/gmd-2016-280-AC1-supplement.pdf

**Supplement:**

**A multi-level canopy radiative transfer scheme for ORCHIDEE ( revision r2566), based on a domain-averaged structure factor**

Matthew J. McGrath[1,*], James Ryder[1,*], Bernard Pinty[2], Juliane Otto[1,3], Kim Naudts[1,4], Aude Valade[5], Yiying Chen[1,6], James Weedon[7], and Sebastiaan Luyssaert[1,8]

[1]Laboratoire des Sciences du Climat et de l'Environnement, LSCE/IPSL, CEA-CNRS-UVSQ, Université Paris-Saclay, 91191 Gif-sur-Yvette, France
[2]Land Resources Management Unit, Institute for Environment and Sustainability, Joint Research Centre, European Commission, Ispra, Italy
[3]now at: Climate Service Center Germany, Helmholtz-Zentrum Geesthacht, 20095 Hamburg
[4]now at: Max Planck Institute for Meteorology, Hamburg, Germany
[5]Institut Pierre Simon Laplace, Place Jussieu 4, 75010 Paris, France
[6]now at: Research Center for Environmental Changes (RCEC), Academia Sinica, Taipei, Taiwan
[7]VU Amsterdam, Department of Ecological Science, 1081 HV Amsterdam, Netherlands
[8]now at: VU Amsterdam, Department of Ecological Science, 1081 HV Amsterdam, Netherlands
[*]Equal contributions

*Correspondence to:* James Ryder
(james@jamesryder.org)

**Abstract.** In order to better simulate heat fluxes over multilayer ecosystems,  for example, tropical forests and savannahs, the next generation of Earth system models will likely include vertically-resolved vegetation structure and multi-level energy budgets. We present here a multi-level radiation flux transfer scheme which is capable of being used in conjunction with such methods. It is based on a previously established scheme which encapsulates the three dimensional nature of canopies, through the use of a domain-averaged structure factor, referred to here as the effective leaf area index. The radiation fluxes are tracked throughout the canopy in an iterative fashion until they escape into the atmosphere or are absorbed by the canopy or soil; this approach explicitly includes multiple scattering between the canopy layers. A series of tests show that the results from the two-layer case are in acceptable agreement with those from the single layer, although the computational cost is necessarily increased due to the iterations. For the 2 level case fewer than 15% of results across the variable range have a divergence greater than 0.01, fewer than 2% greater than 0.02, and none have a divergence greater than 0.03. The ten-layer case is less precise, but still provides results to within an acceptable range (40% of results across the variable range have a divergence greater than 0.01, 10% across the variable range have a divergence greater than 0.03, though just 1.5% greater than 0.05). This new approach allows for the calculation of radiation flux transfer in vertically resolved vegetation canopies as simulated in global circulation models.

**1 Introduction**

The  present generation of land-surface models fall broadly into two categories. Detailed models exist which provide a close simulation of interactions at a local scale, for example the CANVEG model (Baldocchi et al., 2002) . Such models may include multiple canopy levels and state-of-the-art parameterisation of transport, hydrology and the vegetation-atmosphere interface at the leaf level. The other category are empirical models, which are more suited to regional studies or use as part of global earth system models. This is a non-physical representation in which either the canopy, or often the canopy and the soil surface combined, are simulated as a single object (the 'big-leaf' ). In this case canopy processes are expressed by means of an aggregate expression that encapsulates or summarises the whole canopy. Such a scheme may show bias to conserve in particularly the quantity for which it was designed (*e.g.*, either transpiration rate or $CO_2$ flux in the case of stomatal conductance (McNaughton, 1994) ). Accurate representations of photosynthesis and energy transfer within the canopy are therefore principal reasons for using multilayer canopy models both for single site assessments and for regional and global studies within large-scale land surface models.

The strengths and weaknesses of these two different approaches to canopy modelling have been long discussed, for example by Raupach and Finnigan (1988) , who suggest that it is the length scale of the simulation that is the determining factor when making the choice between single layer or multilayer models. For large scale studies a single-layer model may well be sufficient whereas multi-layer models are more appropriate for studies on a smaller scale, or that involve in-canopy processes. The main assumption behind the 'big-leaf' model is that a single value of leaf physiological properties can be found which adequately represents the entire canopy; given the non-linear response of photosynthesis to incoming light intensity (Björkmann, 1981), it is difficult to find such a value that works under all conditions  (Leuning et al., 1995b) . In addition, leaf properties change across the canopy, in particular in response to depth and, consequently, the amount of light reaching the leaf. The microclimate differs significantly across the canopy as well, and  varies photosynthetic parameters show  varying responses to temperature (De Pury and Farquhar, 1997). Leaf temperature is a significant driving factor in intra-canopy radiation fluxes (Zhao and Qualls, 2005) and chemical processes, such as the emission of biogenic volatile organic  compounds (Guenther et al., 1993).

 In fact, temperature gradients within canopies are often significant, even within short grassland stands (Qualls and Yates, 2001) . Studies that model the energy budget using an in-canopy profile - which also requires a multi-layer radiation model - found that an accurate representation of in-canopy temperature profiles was important for the simulation of sensible heat fluxes. In one study in particular, ignoring such gradients resulted in sensible heat fluxes that were 61% of reference measured values (Baldocchi and Wilson, 2001) . Other studies (Ogée et al., 2003; Carrer et a found that simulations of the energy budget were improved with the simulation of profiles. A multilayer model is required to accurately represent these gradients and vertical heterogeneity within the canopy

energy budget and photosynthetic calculations. This suggests using canopylayers which are a function of height within the canopy.

Many previous studies have indicated problems with holding canopy spectral parameters constant as a function of canopy depth (*e.g.*, Lemon et al. (1971); Sinclair et al. (1976); Sellers et al. (1992); De Pury and Farquhar (1997) ), although the difference between multi-level and 'big leaf' models is sometimes acceptable (de Noblet-Ducoudré et al., 1993) . One significant weakness in the 'big leaf' model is the lack of adaptability to changing canopy parameters. For example, a forest management regime can reduce the total amount of leaf area in a stand by removing smaller trees. A 'big leaf' model represents this change by a reduction in Leaf Area Index (LAI), while a more detailed multilayer model can also change the distribution of the LAI in the canopy, which can have significant effects on the budget of radiation flux.

A further issue is the the lack of detail of canopy models running on a regional or global scale, compared to the site based studies, for which information about canopy height and profile is measured. There has recently been an improvement in measurement capability within canopies, as opposed to an alternative approach, such as layers based on the density of LAI (Leaf Area Index ) . The same shortwave radiation absorbed for photosynthesis should be used in the development of a more portable LIDAR (Hosoi and Omasa, 2007) , the increased availability (albeit gradually) of more detailed in-canopy datasets of canopy structure and the improved coordination and collation of time series measurements of in-canopy temperature, humidity and trace gases (Chen et al., 2015; Sellers et al., 1997) . The most recent generation of land surface models models include the simulation of canopy growth, profile information and leaf level resistances, such as stomatal conductance (e.g. ORCHIDEE-CAN; Organising Carbon and Hydrology In Dynamic EcosystEms - CANopy; Naudts et al. (2015) ). This means that profile based models will increasingly couple not only with the atmosphere, but with other modules that provide more detail of canopy composition.

Furthermore, the evolution of leaf density in canopy profiles links directly to the simulation of $CO_2$ fluxes within the canopy, by means of the calculation of the energy budget, which requires the creation of a radiation transfer (RT) scheme capable of determining radiation fluxes as a function of height within the canopystomatal conductance. Interactions between leaf irradiance at different parts of the canopy and water stress impact on $CO_2$ concentration in these leaves. For example (Bonan et al., 2012) , a multi-layer model, similar to CANVEG was used together with the SPA (Soil-Plant-Atmosphere) model Williams et al. (1996); Williams (2005) to provide for an improved stomatal conductance simulation. When using a multi-layer canopy model, errors in photosynthesis were considerably reduced, as a result of improved distribution of radiation flux and, in a later work, or improved leaf moisture gradients (Bonan et al., 2014) , when compared to a sun-shade model. As a next step, the development of models such as these enables a consistent approach to canopy modelling that can link energy budget and $CO_2$ models with other land surface model processes.

Radiation flux Transfer (RT) models, that resolve explicitly multiple canopy layers as part of a large-scale land surface model, have already been developed. Multilayer RT models already exist in the literature (Haverd et al., 2012; Carrer et al., 2013). The CanSPART Canopy Semi-analytic Pgap and Radiative Transfer (CanSPART) multilayer model includes a gap fraction calculation that is based on geometric properties of the canopy instead of using a generic clumping factor (Haverd et al., 2012). The multi-level multilevel solution is generated by solving a matrix equation for the which calculates the radiation

fluxes between each level. Carrer et al. (2013) adopted a slightly different approach. The calculation of the  transmission of radiation fluxes to each level (and, by extension, the radiation flux absorbed by each level) is given directly as a product of all higher levels. Hanan (2001), Yanagi and Costa (2011) and Yuan et al. (2014) develop non-iterative two and three layer RT schemes which,  whilst efficient, cannot easily extended to an arbitrary number of layers. Bonan et al. (2014) uses an approach based on CANVEG, mentioned above.

So, as outlined above, accurate representations of photosynthesis and energy transfer within the canopy are the principal reasons for using multilayer canopy models both for single site assessments and for regional and global studies within larger land surface models (Ryder et al., 2016) . To maintain model consistency, layer dimensions should coincide for the energy budget and photosynthetic calculations. This suggests using canopy layers which are a function of height within the canopy, as opposed to an alternative approach, such as layers based on the density of LAI. The same shortwave radiation flux absorbed for photosynthesis should be used in the calculation of the energy budget, which requires the creation of an RT scheme capable of determining radiation fluxes as a function of height within the canopy.

The aim of this paper is to establish how the multilevel model performs compared to a single level model. This information is essential to decide when incorporated into a large scale land surface models, the increase in computational costs and potential loss in precision are worth the increased flexibility for more complicated representations of canopy structure. This objective allows for continued development of this model as a canopy radiation transfer model in land surface models. While it may be of enormous value to the community to identify a subset of parameter space in which an increase in model levels improves the skill of this model, the subset for which it roughly matches the performance of the single level scheme, and that for which performance is impaired (relative to observations), that work goes beyond the scope of this paper.

**2  Background to model development**

The ORCHIDEE-CAN model (Naudts et al., 2015) is a  land surface model that builds on the ORCHIDEE model (Krinner et al., 2005; Bellassen et al., 2010) to simulate the biochemical and biophysical effects related to forest management. To better simulate complex canopies, it includes an optional multi-layer energy budget, which links to other model features such as the simulation of canopy growth, a mix of vegetation types with varying height characteristics on the same pixel, a carbon allocation scheme based on allometric relationships, inhomogenous horizontal distribution (such as tree clumping and canopy gaps), the calculation of leaf-layer resistances across the profile, and a new in-canopy radiation flux scheme. This latter aspect necessitates the new albedo scheme that is described here.

The  scheme should satisfy the following criteria:

1. To be consistent with the vertical canopy structure - ~~which requires a multi-layer radiation model - found that an accurate representation of in-canopy temperature profiles were important for the simulation of sensible heat fluxes. In one study in particular, ignoring such gradients resulted in sensible heat fluxes that were 61of reference measured values (Baldocchi and Wilson, 2001) . Other studies (Ogée et al., 2003; Carrer et al., 2013) also found that simulations of the energy budget were improved with the simulation of vertical profiles.~~

    i.e. the leaf area index profile, that is used across all parts the model, the  albedo profile should  be calculated using the same technique as that used to parameterise gaps in the canopy, which in ORCHIDEE-CAN is through use of the Pgap model (Haverd et al., 2012). This assigns a statistical distribution of trees of varying heights and sizes.

2. The scheme should also be flexible enough to be applied across a broad range of vegetation types.

3. The scheme will need to be parameterised at the global scale, a task that will be facilitated if proper global satellite products are available. From this point of view, it makes sense to use a 1-D RT scheme which has been designed to recover and use information from 3-D remote sensing products in a consistent way.

4. The numerical solution of the scheme should be suitable for implementation into a multi-level simulation  to make full use of the advantages of the profile approach that are detailed above.

5. The increase in computation time should be acceptable. In state-of-the-art coupled atmospheric-land models, the atmospheric model requires about 10 times as much computer resources than the land surface model (this is approximately the ratio that applies to the ORCHIDEE model for the uncoupled versus the version coupled to the LMDZ (Laboratoire de Météorologie Dynamique Zoom) atmospheric model). The albedo model should thus not increase the computational cost by a factor of two or more to avoid the land surface becoming a rate limiting step in coupled atmospheric-land models.

~~The rest of this paper is organised as follows. The next section explains the essentials of the RT model proposed by Pinty et al. (2006) , and extends it to multiple layers. Details of the algorithm developed according to this theory are also presented. Results are given comparing the two-layer and ten-layer scheme directly to the one-layer scheme, applying parameter values which are representative of realistic environmental conditions (n. b. the 'REAL' parameter set), together with those spanning more of parameter space (n.b. the 'ALL' paramter set, which includes a known pathological case). Finally, we present the limitationsalbedo model and summarise its performance against the single layer case~~ radiation flux transfer scheme discussed here.

**3 Theory**

The one-layer scheme is described in detail by Pinty et al. (2006). The single layer albedo scheme in that paper was extensively benchmarked against three dimensional Monte Carlo simulations. A comparison was also conducted against a complex range of three dimensional scenario in the context of the RAdiation Model Inter-comparison (RAMI) (Widlowski et al., 2011) The single layer albedo scheme is able to fit closely any situation irrespective of the structural and radiative properties, and we are therefore justified in assessing performances of the multiple layer albedo scheme here against the extensively validated single layer model.

We here use the same notation and terms as developed in Pinty et al. (2006), for consistency. As in that paper, the term 'background' used here refers to all elements except for the vegetation - essentially the soil layer, snow and leaf litter. Briefly, this scheme computes the absorption, transmission, and scattering of incoming radiation fluxes by vegetation canopies by considering three interactions, as follows:

1. The first interaction is the radiation flux that does not collide with the canopy vegetation at all, and reflects off the background and back into the sky, with no interception by vegetation (this is 'black canopy radiation', or radiation flux with no contributions from canopy interactions; $R_{bgd}^{uncoll}$).

2. The second is the radiation  flux that collides with the canopy elements, with a probability to be transmitted through, absorbed by and scattered by the canopy with no contribution from the background (this is 'black background radiation'); $R_{veg}^{coll}$).

3. The third term consists of all radiation flux which collides with both the canopy and the background, before being scattered into the atmosphere ('multiple interaction terms'; $R_{bgd}^{coll}$; light reflected following multiple collisions between vegetation and background).

From here, one can consider each of the three possible fates of a  stream of radiation flux entering a given canopy layer from the top:

1. It can be transmitted through the layer without colliding with any vegetative elements ($T_{veg}^{UnColl}$).

2. It can be transmitted through the bottom of the layer after striking vegetation one or multiple times ($T_{veg}^{Coll}$).

3. It can exit through the top of the layer (effectively 'reflected') after colliding with vegetation one or multiple times ($R_{veg}^{Coll}$).

These three possibilities are illustrated in the top panel of  figure 1, and they constitute the basic step in the multi-level approach. Conceptually, this picture is directly comparable to the original model of Pinty et al. (2006) for the case of a non-reflecting background.

The extension of this approach to the multi-level case is conceptually straightforward, although the implementation requires some modifications. A single unit of radiation flux (originating from either a direct or diffuse  source) is projected

into the top layer of the canopy. The probability that this unit will follow one of the three paths in the top panel of Fig. 1 is computed by solving the equations of Pinty et al. (2006) for the top layer of the canopy assuming a black background (so that at this step no radiation flux enters the layer from below). The estimate of the various radiation fluxes requires knowledge about the effective values of the effective Leaf Area Index ($LAI_{eff}$), single scattering albedo ($w_l$) and scattering phase function for that particular layer. These and other key variables used herein are listed in Table 1, but, for readability, are defined below:

- collimated $LAI_{eff}$: this represents the Plant Area Index (i.e. taking into account leaves and trunk) averaged over light incident from all angles, and taking into account canopy gaps.

- leaf single scattering albedo ($w_l$): the fraction of light that is reflected, as assigned to individual leaves

- background reflectance ($R_{bgd}$): the fraction of light that is reflected from the surface and background elements below the dominant tree canopy

- leaf forward scattering efficiency ($d_l$): the fraction of light that is scattered onwards by individual leaves

- solar zenith angle ($SZA$): the angle between the zenith and the solar position at any given time

For consistency, we will use '$LAI_{eff}$', to also denote LAI in discussing the sensitivity studies. As we are not using LAI from data there is no reason to convert to the $LAI_{eff}$ required by the model.

The fraction scattered off the top layer and back into the atmosphere will not have another chance to interact with the canopy, and therefore it becomes the first approximation to the top of canopy albedo. The two fluxes which radiation fluxes that exit the bottom of the layer are used as inputs into the layer underneath, together with radiation fluxes entering that layer from below (i.e. from the ground, in the two-layer model). The fate of these radiation fluxes is calculated by again solving the equations of Pinty et al. (2006) for this layer, and the resulting radiation fluxes are followed until all of the original incoming radiation flux has either been absorbed by the background, absorbed by the canopy, or reflected back into the atmosphere. The background is treated as a further layer, with the difference that radiation flux can only be absorbed or reflected by it, and the proportion of radiation flux reflected is simply proportional to the background reflectance.

Given that the radiation flux transfer problem is solved using a two-stream solution for each individual sub-layer, our proposed scheme assumes that the exiting radiances in the upward and downwards directions can be approximated approximated by the exiting radiation fluxes - i.e. directionality is not maintained. This assumption becomes especially critical for layers of intermediate density. The intensity of the diffuse radiation fluxes is numerically small for low vegetation density conditions (i.e. with an $LAI_{eff}$ that approaches 1unity), as a single scattering single-scattering albedo regime dominates.

The assignment of an effective LAI $LAI_{eff}$ to each level is an appropriate simulation assumption as the direct transmission values of the light transmitted can be calculated as the product of the layers concerned. However, we make the assumption that other factors, such as the single scattering albedo, can be assigned directly. In fact these values will be affected by back-scattering and diffuse transmission of light between layers, and so depend slightly on the effective LAI $LAI_{eff}$ at each level. A refinement of the model, beyond the scope for this paper, would be to run a series of convergence tests to determine more accurately the single-scatter single-scattering albedo for each level.

One important consideration in the original scheme of Pinty et al. (2006) is that of the radiation  that is transmitted through the canopy originating from a diffuse source, without colliding with the vegetation. Several versions of this formulation were given (Eqns. 16, 18, and 19 in Pinty et al. (2006), taking into account a small error, in the original version, of Eqn. 16). The most accurate solution is Eqn. 16 from Pinty et al. (2006):

$$\overline{T_{\text{veg}}^{\text{UnColl}}} = \exp\{-LAI_{eff}/2\}\left(1 - (LAI_{eff}/2) + (LAI_{eff}/2)^2 \times \Gamma(0, (LAI_{eff}/2))\right) \tag{1}$$

 For more details on Eqns. 1, 2, and 3 readers are referred to Pinty et al. (2006). For all three equations the abbreviations are defined in Table 1.

Given the inclusion of the incomplete gamma function, this equation is computationally somewhat expensive to solve. Therefore Pinty et al. (2006) proposed two approximate solutions, as well, noting that the following works fairly well for typical values of the  $LAI_{eff}$:

$$\overline{T_{\text{veg}}^{\text{UnColl}}} = \exp\{-LAI_{eff}/2\}\left(\frac{1}{1 + LAI_{eff}/2}\right) \tag{2}$$

Assuming the argument of the exponential is small enough, even this approximation can be further simplified, weighting by an empirical factor (the determination of which is explained in Section 2.5 of Pinty et al. (2006)):

$$\overline{T_{\text{veg}}^{\text{UnColl}}} = \exp\{-0.705 * LAI_{eff}\} \tag{3}$$

The impact of these three equations for the un-collided  diffuse source radiation flux on the multi-level solution will be explored in more detail in the following section.

**4   Algorithm**

~~As can be understood from Figure 1, the number of fluxes increases exponentially with each step. It was anticipated that in truly pathological cases (i.e. those with a highly reflective background and non-absorbing canopy elements) the scheme would take dozens of steps to converge. In order to avoid the difficulty of tracking each flux at every step, the fluxes were combined. It is clear from the theoretical description thatthe directionif theisotropicthat is to say,~~ i.e. diffusely reflecting) and our canopy elements are bi-Lambertian (Lambertian for both transmitted and reflected light), we do not have to track an upward collimated radiation flux; any radiation flux travelling in the upward direction is necessarily scattered, and therefore will be diffuse according to our assumptions of the scattering elements. This results in only three radiation fluxes to track for each layer, as illustrated - $T_{coll}$ (transmitted collided), $T_{uncoll}$ (transmitted uncolled) and $R_{coll}$ (reflected uncolled).

As is outlined in Figure 1, the number of radiation fluxes increases exponentially with each step. It was anticipated that in truly pathological cases (i.e. those with a highly reflective background and non-absorbing canopy elements) the scheme would

take dozens of steps to converge. In order to avoid the difficulty of tracking each radiation flux at every step, the radiation fluxes were combined. The algorithm is outlined below:

1. Use the one-layer model of Pinty et al. (2006) to compute the fraction of radiation flux which is reflected, transmitted after scattering, and transmitted without interacting with the canopy for each layer for both collimated and  diffuse radiation flux sources, assuming an input radiation flux of unity. These are referred to as the "unscaled" radiation fluxes.

2. Initialise all the radiation fluxes. For a collimated radiation flux source, the atmospheric collimated downwelling radiation flux is set equal to unity. For  a diffuse source of radiation flux, the atmospheric  diffuse downwelling radiation flux is also set equal to unity.

3. Begin the convergence loop.

4. Initialise the variables which track the  radiation fluxes that are generated by this step (referred as the radiation fluxes for the next step).

5. Start the loop over all levels in the system.

6. For each layer, determine the fraction of collimated downwelling radiation flux for the layer which is converted into downwelling collimated radiation flux for the lower layer (i.e. no interaction with the canopy), downwelling  diffuse radiation flux for the lower layer (forward scattering by canopy elements), and upwelling  diffuse radiation flux for the upper layer (back scattering by canopy elements). This consists of multiplying the current radiation flux for the layer by the unscaled  radiation fluxes that are computed above.

7. Determine the fraction of  diffuse downwelling radiation flux for this layer which is converted into downwelling  diffuse radiation flux for the lower layer (no interaction with the canopy), downwelling  diffuse radiation flux for the lower layer (forward scattering by canopy elements), and upwelling  diffuse radiation flux for the upper layer (back scattering by canopy elements). Note that no downwelling collimated radiation flux can be produced by this step.

8. Determine the fraction of  diffuse upwelling radiation flux for this layer which is converted into upwelling  diffuse radiation flux for the upper layer (no interaction with the canopy), upwelling  diffuse radiation flux for the upper layer (forward scattering by canopy elements), and downwelling  diffuse radiation flux for the upper layer (back scattering by canopy elements). Note that no downwelling collimated radiation flux can be produced by this step.

9. Any downwelling collimated or  diffuse radiation flux which reaches the background can become upwelling  diffuse radiation flux for the next step by reflecting off it. The background reflectance is a fixed parameter, as in the single layer solution of Pinty et al. (2006).

10. Convergence is satisfied when all radiation fluxes at this step are less than a specified threshold.

 As determined by the algorithm, we can tell that the total top of the canopy albedo is simply the sum of all the upwelling  diffuse radiation flux in the atmospheric layer over all the iteration steps. One can compute the total amount of radiation flux reaching the background in a similar manner, as well as the radiation fluxes within the canopy. The absorption of radiation flux by each canopy layer is calculated by taking the difference of the incoming radiation flux to each layer (from above and below) and the outgoing radiation flux in each direction. For the top layer, this is given by:

$$A_{veg,\,z=z_{top}} = 1 + R^{coll}_{veg,\,z=z_{bottom}} - (R^{coll}_{veg,\,z=z_{top}} + T^{tot}_{veg,\,z=z_{top}}) \tag{4}$$

where $R^{(coll)}_{veg,z=z_{bottom}}$  is the reflected radiation flux from the bottom layer (the sum of all upwelling radiation fluxes), $R^{(coll)}_{veg,z=z_{top}}$  is the reflected radiation flux from the top layer (the top of the canopy albedo), and $T^{tot}_{veg,z=z_{top}}$ is the total radiation flux that is transmitted through the top layer (the sum of all downwelling radiation fluxes from the top layer).

Similarly, the absorption of the bottom canopy layer is given by:

$$A_{veg,\,z=z_{bottom}} = 1 + T^{tot}_{veg,\,z=z_{bottom}} * R_{bgd} - (R^{coll}_{veg,\,z=z_{bottom}} + T^{tot}_{veg,\,z=z_{bottom}}) \tag{5}$$

where $T^{tot}_{veg,z=z_{bottom}}$ is the total radiation flux that is transmitted through the bottom layer, $R_{bgd}$ is the background reflectance $R^{coll}_{veg,z=z_{bottom}}$ is the total upwelling radiation flux from the bottom layer, and $T^{tot}_{veg,z=z_{bottom}}$ is the total downwelling radiation flux from the bottom layer.

The multi-level case requires an iteration scheme yielding an update of the upper and lower boundary conditions associated with each layer until reaching the appropriate radiation flux balance.

**5 Validation**

This new albedo simulation has been written as part of a new albedo module in ORCHIDEE-CAN (Naudts et al., 2015), and integrates with the multi-level calculations of stomatal conductance, and the multi-level energy budget.  The specific number of layers to be used depends on the other aspects of the canopy model, particularly the energy budget and the photosynthesis scheme. It is the nature of these latter schemes that would determine the number of layers to be used. For example, an energy budget calibrated for an understorey/overstorey ecosystem may run on two layers, whilst an energy budget for a more complex canopy might use ten. The number of layers in the albedo scheme would match those required by the other processes; our scheme is sufficiently flexible that changing this in the model is straightforward. For the set of tests in this study, the model is run for 1, 2 and 10 levels.

For the set of tests in this study, the model is run with six independent variables, that are fed directly into the albedo routine in order to access the capability of this scheme. All simulations were retained: 5,000 pixels, 12 plant functional types per pixel, 17,520 albedo calculations per year per plant functional type. Over one billion albedo calculations per simulated per year, with

no hard constraints on some of the parameters. To make sure that our simulations stay within check it is important to check how the model behaves outside the expected range of parameters, this represents a sanity check of the model and its implementation. For large scale simulations this makes the difference between a good and an excellent model. Even rare case that happen once in a million tests are rather frequent for the applications of earth system modelling

5   It could be that by assuming a uniform distribution we have overly distorted the results in favour of the input values on the 'tails' of a normal distribution, when performing the tests otherwise would have shown a larger percentage passing the 0.01 difference threshold. However, it remains challenging to identify a strong and robust basis to weigh some particular combination of parameters and accordingly with adopted a uniform distribution to cover a large range of conditions.

Five of the variables also influence the single-layer scheme: the  values of the total
10  $LAI_{eff}$, the single-scattering albedo ($w_1$), the forward scattering efficiency ($d_1$), the solar zenith angle (SZA - n.b. a value of $0°$ corresponds to the sun directly overhead), and the background reflectance. In addition to the single layer case, we must also look at the distribution of the  $LAI_{eff}$ between the two canopy layers.  As six independent variables is too many to perform
15 an exhaustive sensitivity analysis, we selected just two parameter sets. One parameter set, denoted $ALL$, covers a wide range of possible values of the parameters. The second set, denoted $REAL$, focuses on a range of values which are more likely to be observed in nature.  $REAL$ is a subset of $ALL$, comprising almost an order of magnitude fewer points. The specific values used are given in Table 2.

For each possible combination of parameters given in Table 2, we computed the single- and two-layer solutions. The sin-
20 gle layer case has been extensively validated (Pinty et al., 2006), and therefore we consider it to be a good reference case  . If the multi-level results match the single layer results, they can be considered as acceptable for further applications. There are four major output radiation fluxes of interest: the top of the canopy albedo ($R_{veg,\ z=z_{top}}^{coll}$), the total transmission through the canopy ($T_{veg}^{total}$), the total absorption by the canopy ($A_{veg}^{total}$), and the total absorption by the background ($A_{background}^{total}$). All of these radiation fluxes are present in both the single and multi-level schemes, which makes them
25 easy to compare, even if slightly more work is required in the multi-level case. The total canopy absorption is given by:

$$A_{veg}^{total} = A_{veg,z=z_{top}} + A_{veg,z=z_{bottom}} \tag{6}$$

while the absorption of the background is whatever radiation flux is not reflected  or absorbed by the canopy:

$$A_{background}^{total} = 1 - (A_{veg}^{tot} + R_{veg,z=z_{top}}^{coll}) \tag{7}$$

For consistency when comparing results, the following set of thresholds was set for determining the level of agreement
30 between the albedo that was calculated, between the single-level and the multi-level model:

– a difference of $< 0.01$ represents a minimal change, which would be equivalent to a top of canopy measurement error

– a difference of 0.01 - 0.05 represents an acceptable change

– a difference of > 0.05 represents a substantial change anything above this approaches the differences between the albedo for different land surface types (e.g. evergreen versus deciduous forest) and so is ruled out

In order to investigate whether some parameter settings are more prone to deviations than others, an approach of Generalised Additive ModelS (GAMS, Hastie and Tibshirani (1990)) was applied to both sets of model output -  across the *REAL* and the *ALL* value range. The approach calculates the extent to which the variance of each of the two dependent variables can be explained by each of the input terms in the model.

First of all, the full model variance is calculated for each of the  dependent variables - in terms of all independent input variables, both for the first order, and for the second order, tensor, interactions. Next, the full model variance calculation is conducted again, but with one term excluded - this calculation is repeated for each of the first order and second order terms. The difference between the full model variance and the variance with one term excluded, as a fraction of the null model variance provides a value for the contribution to the variance from each term.

**6 Results**

In assessing the performance of the model, our aim is to implement procedures that replace statistical subroutines with more physically realistic processes. Different land surface models and research groups have different requirements for the accuracy of the models that they use, as the coupling of these models are complex, non-linear functions. Therefore we only report deviations from the single layer model here, and potential users should make their own judgements regarding the performance of the model in their schemes, given these deviations.

As outlined in Table 4, the Root Mean Square Error (RMSE) falls below $1.0e^{-2}$ for the four main radiation fluxes. The four main radiation fluxes of the two-layer case thus agree well with those for the single layer case for the  *REAL* parameter set (Figure 2). The results are encouraging as there seem to be no obvious cases where the multi-level approach fails, although deviations occur more frequently for radiation fluxes which are not extreme (0.2–0.4).  Comparable results are observed for the *ALL* parameter set (Figure  S1).

The difference between the one- and two-layer radiation fluxes depend strongly on the  radiation flux of interest (Figure 3). The two-layer radiation flux for the transmission through the canopy is larger than that in the one-layer case. The magnitude of the difference can reach  0.02, although >95% of values are well below that. A small fraction of the differences between the two-level and the one-level case may be related to the assumption of an assigned effective  single-scattering albedo to each level in the model, as discussed in Section 2. These observations for the  *REAL* parameter set appear to be also true for the  *ALL* parameter sets (Figure S2).

As the fraction of  $LAI_{eff}$ found in each layer cannot be specified in the single-layer case, it is instructive to see how the variation of this quantity effects the agreement between the two models (Figure 4). The magnitude

of the differences is identical to that in Figure 3, which is to be expected as the data sets are identical. It is also unsurprising that when all of the  $LAI_{eff}$ is in either the upper or lower layer the two-layer radiation fluxes match the one-layer radiation fluxes; in these cases, the model reduces to the single layer version. One interesting observation from Figure 4 is that the

5   differences are asymmetric - a 1:9 distribution of top:bottom  $LAI_{eff}$ gives greater differences than a 9:1 distribution. This suggests that the initial scattering by the top layer is more influential than multiple-scattering between the background and bottom layer. Again, these observations for the  *REAL* parameter set appear to be also true for the  *ALL* parameter sets (Figure S3).

    Figures 2, 3, and 4)

10   demonstrate the visual extent of variation in the calculated albedo between the single and two-level modelt. To quantify model performance, we computed the fraction of simulations for which the difference between the one- and the two- layer case, not distinguishing between radiation fluxes, was larger than a specified threshold (Table    3). For the *REAL* dataset using the two layer model, the fluxes of the majority of

15   test points differ by  0.01   or less compared to the well-validated single-layer model in 85% of cases, and by 0.02 or less in 98% of cases (see Table 3). There are no incidences for which the comparisons differ by more than 0.03. In this study we assess the performance of the multi-layer albedo model by applying thresholds to performance against the single layer case. It is beyond the scope of

20   this work to evaluate the global performance of the multi-level albedo scheme within the land surface model.

    At first glance, it may seem  counter intuitive in Table 3 that the numbers are higher for the  *REAL* parameter set than the  *ALL* parameter set, as  *REAL* should be a subset of *ALL*. The reason for this is that the numbers give the fraction of points greater than a threshold, not the total number of points. For example,  *ALL* contains 224,575 simulations which differ by more than 0.01 from the single-layer solution, while  *REAL* has only 43,800. This

25   is expected as  *REAL* is a subset of *ALL*. However,  *ALL* has 2,594,592 total simulations while  *REAL* only has 304,128. Therefore, the fraction of divergent points is   greater.

    One natural question arising from Table  3 regards the simulations that differ by more than 0.01. Is there a common trend there which can be identified? To identify possible trends, we isolated all such simulations and applied  the Generalised Ad-

30   ditive Models (GAMS), as described in the previous section. Figure 5 depicts the resulting calculated deviances for ($\alpha_{collim}^{multi}$ - $\alpha_{collim}^{single}$), which is the difference in calculated collimated albedo between the multi-layer and single layer model. The analysis reveals that, for the  *REAL* parameter set, medium values of LAI, small forward scattering efficiencies, evenly distributed vegetation, and high  single-scattering albedos all lead to increased frequencies of points which deviate significantly from the single layer model. The solar zenith angle and the background reflectance appear to have little effect on the

35   differences between the one- and two- layer models.

Figure 6 demonstrates the relevant trend in $n_{steps}$, which is demonstrated to depend heavily on $w_l$, the single-scattering albedo.

The complete model is run over two layers, as in the previous tests, and also over ten layers. We found that the most significant contribution to the variance for both the ten- and two- layer cases comes from $w_l$, the  single-scattering albedo. As the layers in the canopy profile have been added, this demonstrates that factors that control the passage of fluxes between layers as expected have the most significant influence on the total flux from the canopy.

Figure  7 plots the marginal effects  the effects of each of the independent input values against the output variable (which is in this case the outcome of the radiation flux difference threshold test). Note that for a threshold of 0.01, the ten- layer case requires almost 10 times as many simulations than the two layer case to fail the threshold. The number of simulations that fail the threshold for the ten layer case equal the number of simulations of the two layer case when the threshold is increased from 0.01 to 0.03.

From a computational point of view,  the iterative procedure introduced above will be more demanding of computer resources than the original one-layer scheme. Two valid questions are therefore how much more expensive  could it be, and under what conditions will this expense be increased? To answer these, we have applied a GAMS to the mean number of iteration steps ($n_{steps}$) to convergence. The partitioning of the  $LAI_{eff}$ into the layers also has a significant effect on the expense of the algorithm, requiring more time as the bottom layer becomes empty. The  single-scattering albedo ($w_l$), is the dominant input variable, followed this time by interactions between that variable and $R_{bgd}$, the collimated background reflectance, as shown in Figure 6. The solar zenith angle is not an important factor, resulting in only small differences. These analyses show that as more light is scattered, for example, with a more reflective background or more reflective leaves, the number of steps required to converge can increase by a factor of five (shown in Figure 7). Finally, Figure  8 plots the marginal effects for a difference in albedo between the two- and one- layer model, for the both the  *REAL* and *ALL* datasets. Figure  9 is the corresponding graph for the number of iteration steps ($n_{steps}$), which shows little difference between the two value tests for this metric.

**7  Discussion**

As the multi-level radiation flux transfer model is designed to run at a global scale, it is important to assess the scheme in regions where changes in albedo have the most consequence for the global climate. For example in the northern latitudes albedo has a key effect in the springtime, when solar angles are increasing and there is still a broad snow cover. In these circumstances, the results (Figure  7 shows that as the solar zenith angle reaches a large value, the difference between the calculated albedo, both for a single and the multi-level version, decreases and so the accuracy improves with the seasonal variation in radiation flux. In absolute terms, the calculated albedo has the greatest effect in the tropics, as the largest amount of direct shortwave radiation  flux falls in this region. Of course, the solar zenith angle will have a higher mean value towards

the lower end of the range displayed in figure S4e). So deviations in absolute terms are expected to be relatively low due to the high  $LAI_{eff}$ in the tropics.

The interactions between leaf irradiance in different parts of the canopy and water stress have an impact on $CO_2$ concentration within these leaves. When using a multi-layer canopy model, errors in  photosynthesis were considerably reduced, as a result of improved radiation flux distribution (Bonan et al., 2012) and, in a later work, also by improved leaf moisture gradients (Bonan et al., 2014), when compared to a sun-shade model. As a next step, the development of models such as these enables a consistent approach to canopy modelling that can link energy budget and $CO_2$ models with other  Land surface model processes.

There has recently been an improvement in measurement capability within canopies, such as the development of a more portable LIDAR (Hosoi and Omasa, 2007), the increased availability (albeit gradually) of more detailed in-canopy datasets of canopy structure and the improved coordination and collation of time series measurements of in-canopy temperature, humidity and trace gases (Chen et al., 2015; Sellers et al., 1997). Following this trend in data availability, the most recent generation of  land surface models include the simulation of canopy growth, vertical canopy profile information and leaf level resistances, such as stomatal conductance (e.g. in ORCHIDEE-CAN (Naudts et al., 2015) and the CLM (Community Land Model) (Bonan et al., 2012, 2014)). This means that profile based models will increasingly couple not only with the atmosphere, but with other modules that provide more detail of canopy composition.

Researchers select the models they include in their simulations based on several factors: computational demand, flexibility, and accuracy being among the most important. From the figures and analysis presented here, the canopy multilevel radiation transfer does not reduce computational demand or improve accuracy. However, it provides flexibility for researchers to include more detailed canopy models in their work. In our view such developments as described here will enable us to start using observational data which in the long run could help to improve the model. For example, calculating isotopic fractionation and mixing will not improve the simulations themselves but it would be a very powerful tool to validate some of the underlying processes. Furthermore, adding more detailed canopies and energy budgets are necessary, if we want to use remote sensed surface temperatures.

**8 Conclusions**

We have developed an algorithm to extend a powerful single-layer canopy radiative transfer model to multiple layers. The original radiative transfer scheme incorporated three dimensional canopy structure through the consideration of a domain-averaged structure factor, providing the two-stream radiation fluxes as output. Our extension here tracks the radiation fluxes as they pass through the canopy layers, using an iterative procedure, until they escape into the atmosphere or the background; in this way, multiple scattering between the canopy layers is taken into account, as well as the multiple scattering between the canopy and the background included in the single-layer scheme.

Despite the fact that computational cost increased and divergence of the multi- compared against the one- layer model occurs, especially for realistic parameter values, the magnitude and sensitivity of the divergence does not hamper use of the model for global simulations.

The results of the tests presented here are encouraging, showing  that deviations between the one- and two-layer models  are no more than 0.04 albedo units. While this difference is not insignificant,  fewer than 15% of divergences are greater than 0.01 in the two level case, and for the ten level case, 10% across the variable range have a divergence greater than 0.03. Some parameters (primarily $w_l$, the leaf  single-scattering albedo, but also $R_{bgd}$, the background reflectance) appeared to have larger effects on the agreement between the two approaches than others. The computational cost of the multi-level approach was also examined. Again, some parameters (such as the background reflectance and the  single-scattering albedo) had a larger impact on the number of steps required to converge to the iterative solution than others, as is identified by the GAMs analysis tests. The number of steps required to converge was only loosely correlated to increased differences between the one- and two-layer results. These results indicate that, while systems with highly reflecting background (like snow) and high  single-scattering albedo values (for example, in the infrared band) may lead to large differences with the single-layer case, overall this scheme is robust and serves as a powerful  step towards the next generation of global  land surface models with multi-level energy budgets and  three dimensional vegetation structure.

**9  Code availability**

The ORCHIDEE-CAN code and the LibIGCM run environment are open source and distributed under the CeCILL licence (http://www.cecill.info/index.en.html). Nevertheless readers interested in running ORCHIDEE-CAN (revision 4262) are encouraged to contact the corresponding author for full details and latest bug fixes.

[revised manuscript text omitted]

**Figure 1.** Schematic representation of the first three steps in the multi-level algorithm. The left side of the figure represents the first three steps of the situation where the initial light stream does not collide with vegetation. The right side of the figure represents the first three steps of the situation where the initial light stream does collide with the vegetation. 'T' and 'R' represent packets of light which are transmitted and reflected, respectively. 'coll' represents light that has collided with vegetation elements in the present timestep and 'uncoll' represents light that has not collided with vegetation. 'coll, uncoll' for example represents light uncollided with vegetation in the previous step, that has subsequently collided with vegetation in the present step, and so on for 'coll, coll', 'uncoll, coll' and 'uncoll, uncoll'.

[Figure]

**Figure 2.** The radiation fluxes of the the total transmission through the canopy (TRAN, or $T_{veg}^{total}$ in text), top of the canopy albedo (ALB, or $R_{veg,\ z=z_{top}}^{coll}$ in text), the total absorption by the canopy (CAN ABS, or $A_{veg}^{total}$ in text), and the total absorption by the background -  specifically soil, snow and leaf litter (SOIL ABS, or $A_{background}^{total}$). Figure is for the two layer model ($\phi_2$) as a function of the one layer corresponding results ($\phi_1$) for a  wide range of input parameters (see table 2). Different radiation fluxes are represented by different colours, with each factor highlighted in turn: a) transmitted light; b) light reflected from top of canopy; c) light absorbed by canopy; d) light absorbed by soil. This figure corresponds to the  *REAL* dataset; 1 in every 75 data points plotted for clarity

[Figure]

**Figure 3.** The radiation fluxes of the the total transmission through the canopy (TRAN, or $T_{veg}^{total}$ in text), top of the canopy albedo (ALB, or $R_{veg,\ z=z_{top}}^{coll}$ in text), the total absorption by the canopy (CAN ABS, or $A_{veg}^{total}$ in text), and the total absorption by the background -  specifically soil, snow and leaf litter (SOIL ABS, or $A_{background}^{total}$). Figure shows the signed difference between the two layer results and the one layer results ($\phi_2 - \phi_1$) as a function of the one layer results for a broad range of input parameters. Different radiation fluxes are represented by different colours, with each factor highlighted in turn: a) transmitted light; b) light reflected from top of canopy; c) light absorbed by canopy; d) light absorbed by soil. This figure corresponds to the  *REAL* dataset; 1 in every 75 data points plotted for clarity

[Figure]

**Figure 4.** The radiation fluxes of the the total transmission through the canopy (TRAN, or $T_{veg}^{total}$ in text), top of the canopy albedo (ALB, or $R_{veg,\ z=z_{top}}^{coll}$ in text), the total absorption by the canopy (CAN ABS, or $A_{veg}^{total}$ in text), and the total absorption by the background - that is to say soil, snow and leaf litter (SOIL ABS, or $A_{background}^{total}$). Figure is for the difference in radiation fluxes between the two-layer and the single-layer results ($\phi_2 - \phi_1$) as a function of the fraction of LAI in the top layer for a broad range of input parameters. Different radiation fluxes are represented by different colours, with each factor highlighted in turn: a) transmitted light; b) light reflected from top of canopy; c) light absorbed by canopy; d) light absorbed by soil. This figure corresponds to the  *REAL* dataset; 1 in every 75 data points plotted for clarity

[Figure]

**Figure 5.** Contribution of each input variable to the deviance in the value of $(\alpha_{collim}^{multi} - \alpha_{collim}^{single})$, the difference of the collimated albedo for the multi-level (two layer version in blue and ten layer in red) and for the single layer model. Different radiation fluxes are represented by different colours, with each factor highlighted in turn: a) transmitted light; b) light reflected from top of canopy; c) light absorbed by canopy; d) light absorbed by soil. This figure corresponds to the *REAL* dataset.

[Figure]

**Figure 6.** Contribution of each input variable to the deviance in the value of $n_{steps}$, the total number of iterations required to reach a result in the multi-level model (two layers in blue and ten layers in red). This figure corresponds to the *REAL* dataset.

[Figure]

**Figure 7.** Marginal values of each of the independent input variables against the fraction of points for which the absolute difference of $(\alpha^{multi}_{collim} - \alpha^{single}_{collim})$, is less than 0.01 (red), 0.02 (green) and 0.03 (orange) for the 10 level multi-layer model, and below 0.01 (blue) for the 2 level multilevel model. This figure corresponds to the *REAL* dataset.

[Figure]

**Figure 8.** Marginal values for each of the independent input variables against $n_{steps}$, the total number of iterations required to arrive at a solution, for the *ALL* dataset, for 2 level model (blue) and the 10 level model (red). This figure corresponds to the *REAL* dataset.

[Figure]

**Figure 9.** Marginal values for each of the independent input variables against $n_{steps}$, the total number of iterations required to arrive at a solution for the 2-level case, for the *REAL* dataset (blue), and the *ALL* value dataset (red).

**Table 1.** Key to abbreviations

| Description | Term used in code | Symbol in text |
|---|---|---|
| Independent input variables to model | | |
| collimated effective Leaf Area Index (LAI) | ilaieff | $LAI_{eff}$ |
| fractional effective LAI | laieff_frac | $LAI_{eff}^{frac}$ |
| leaf single-scattering albedo | iwl | $w_l$ |
| background reflectance | irbgd | $R_{bgd}$ |
| leaf forward scattering efficiency | idl | $d_l$ |
| solar zenith angle | itheta | $SZA$ |
| fraction of LAI contained within top half of the canopy | isplit | $f_{LAI,tot}$ |
| Independent output variables to model | | |
| collimated albedo for a single layer | collim_alb_tot_1 | $\alpha_{collim}^{single}$ |
| collimated albedo for total layers | collim_alb_tot | $\alpha_{collim}^{multi}$ |
| number of model iterations | nsteps | $n_{steps}$ |
| Other abbreviations for light packets | | |
| reflected stream; collided with background; uncollided with vegetation | * | $R_{veg}^{uncoll}$ |
| reflected stream; uncollided with background; collided with vegetation | * | $R_{veg}^{coll}$ |
| reflected stream; collided with background; collided with vegetation | * | $R_{bkg}^{coll}$ |
| transmitted stream; uncollided with vegetation | * | $T_{veg}^{uncoll}$ |
| transmitted stream; collided with vegetation | * | $T_{veg}^{coll}$ |
| top canopy level | | $z_{top}$ |
| bottom canopy level | | $z_{bottom}$ |
| range of input values encompassing a wide range of parameters | | $ALL$ |
| range of input values encompassing a smaller, more realistic, range of input parameters | | $REAL$ |

* calculated as sum of components in code (no direct term)

**Table 2.** The range of values used for the  *ALL* test case (top) and the  *REAL* test case (bottom).  $LAI$ is in units of leaf area per square meter of land, $SZA$ is the solar zenith angle in degrees, and the rest of the variables are unitless.

| Variable | Values |
|---|---|
| $LAI$ | (1.0, 2.0, 3.0, 4.0, 5.0, 6.0, 7.0, 9.0, 11.0) |
|  $f_{\text{LAI}_{\text{eff}},\text{top}}$ | (0.0, 0.1, 0.2, 0.3, 0.4, 0.5, 0.6, 0.7, 0.8, 0.9, 1.0) |
| $d_{\text{l}}$ | (0.1, 0.5, 1.0, 1.5, 2.0, 5.0, 10.0) |
| $SZA$ | (0.01, 20.0, 40.0, 60.0, 80.0, ISO) |
| $w_{\text{l}}$ | (0.001, 0.05, 0.1, 0.15, 0.2, 0.3, 0.4, 0.5, 0.6, 0.7, 0.8, 0.9, 0.999) |
| $R_{\text{bgd}}$ | (0.001, 0.05, 0.1, 0.15, 0.2, 0.3, 0.4, 0.5, 0.6, 0.7, 0.8, 0.9, 0.999 ) |
| $LAI$ | (1.0, 2.0, 3.0, 4.0, 5.0, 6.0) |
|  $f_{\text{LAI}_{\text{eff}},\text{top}}$ | (0.0, 0.1, 0.2, 0.3, 0.4, 0.5, 0.6, 0.7, 0.8, 0.9, 1.0) |
| $d_{\text{l}}$ | (1.0, 1.5, 2.0) |
| $SZA$ | (0.01, 20.0, 40.0, 60.0, 80.0, ISO) |
| $w_{\text{l}}$ | (0.05, 0.1, 0.15, 0.2, 0.6, 0.7, 0.8, 0.9) |
| $R_{\text{bgd}}$ | (0.05, 0.1, 0.15, 0.2, 0.6, 0.7, 0.8, 0.9 ) |

**Table 3.** The fraction of data points for each combination of layer and parameter set for which the absolute value of the difference between the one-layer and  two-layer case is greater than the specified threshold. Different equations are used for the transmission factor of uncollided  diffuse radiation fluxes, as described in Section 3.

|  Parameter set | | Errors | | | | |
|---|---|---|---|---|---|---|
| | | 0.001 | 0.002 | 0.005 | 0.01 | 0.02 |
| Eq. 1 | | | | | | |
|  ALL |  0.432 | 0.334 | 0.195 | 0.103 | 0.041 | |
|  REAL |  0.489 | 0.380 | 0.234 | 0.117 | 0.034 | |
| Eq. 2 | | | | | | |
|  ALL |  0.390 | 0.306 | 0.195 | 0.123 | 0.068 | |
|  REAL |  0.457 | 0.378 | 0.251 | 0.145 | 0.063 | |
| Eq. 3 | | | | | | |
|  ALL |  0.461 | 0.359 | 0.200 | 0.084 | 0.010 | |
|  REAL |  0.567 | 0.437 | 0.267 | 0.144 | 0.018 | |

**Table 4.** Root Mean Square Error (RMSE) for comparisons between two layer and one layer models, as plotted in Figure 2

| Collimated light | RMSE |
| --- | --- |
| Two layer absolute versus one layer absolute radiation flux | |
| Transmitted | 6.36e-3 |
| Albedo | 4.21e-3 |
| Absorbed by canopy | 6.28e-3 |
| Absorbed by soil | 3.66e-3 |